



# Harnessing Big Data to Rethink Land Heterogeneity in Earth System Models

Nathaniel W. Chaney[1], Marjolein H. J. Van Huijgevoort[1], Elena Shevliakova[2], Sergey Malyshev[3], Paul C. D. Milly[4], Paul P. G. Gauthier[5], and Benjamin N. Sulman[1]

[1]Program in Atmospheric and Oceanic Sciences,Princeton University, Princeton, New Jersey.
[2]NOAA/Geophysical Fluid Dynamics Laboratory, Princeton, New Jersey
[3]Cooperative Institute for Climate Science, Princeton University, Princeton, New Jersey
[4]U.S. Geological Survey, Princeton, New Jersey
[5]Department of Geosciences, Princeton University, Princeton, New Jersey

*Correspondence to:* N. W. Chaney (nchaney@princeton.edu)

**Abstract.** The continual growth in the availability, detail, and wealth of environmental data provides an invaluable asset to improve the characterization of land heterogeneity in Earth System models—a persistent challenge in macroscale models. However, due to the nature of these data (volume and complexity) and the computational constraints of macroscale models, until now these data have been underutilized for global applications. As a proof of concept, this study explores over a 1/4 degree

(∼25 km) grid cell in southeastern California how to effectively and efficiently harness these data in Earth System models. First, a novel hierarchical multivariate clustering approach (HMC) is used to summarize the high dimensional environmental data space into hydrologically interconnected representative clusters (i.e., tiles). These tiles and their associated properties are then used to parameterize the sub-grid heterogeneity of the Geophysical Fluid Dynamics Laboratory (GFDL) LM4-HB land model. To assess how this data-driven approach to assemble the model tiles impacts the simulated water, energy, and carbon

cycles, model experiments are run using a series of different tile configurations assembled by HMC. The results over the 1/4 degree macroscale grid cell and the underlying 30-meter fine-scale grid in southeastern California show that: 1) the observed similarity over the landscape makes it possible to robustly account for the role of multi-scale heterogeneity in the macroscale states and fluxes with around 300 sub-grid land model tiles; 2) assembling the sub-grid tiles from observed data, at times, leads to noticeable differences in the macroscale water, energy, and carbon cycles; for example, explicit subsurface interactions

between the tiles leads to a dampening of macroscale extremes; 3) connecting the fine-scale grid to the model tiles via HMC enables circumventing the classic scale discrepancies between the macroscale and field-scale estimates; this has potentially significant implications for the evaluation and application of Earth System models.

## 1   Introduction

Spatial heterogeneity plays a critical role in the terrestrial water, energy, and biogeochemical cycles from local to continental

and global scales. This has been recognized for decades in hydrology, ecology, geomorphology, and soil science where it has been observed repeatedly that at multiple temporal and spatial scales, land surface processes have deep ties to an ecosystem's





spatial structure and function. As a result, the macroscale behavior of the water, energy, and biogeochemical cycles cannot be disentangled from their fine-scale processes and interactions (*Beven and Kirby*, 1979; *Wood et al.*, 2011; *Bierkens et al.*, 2014; *Katul et al.*, 2007; *Köppen*, 1936; *Holdridge*, 1947; *Box*, 1981).

Recognizing the importance of multi-scale heterogeneity in the Earth System, in the 1980s and 1990s there was a strong
emphasis to include its role in large scale land surface models (*Beven and Kirby*, 1979; *Avissar and Pielke*, 1989; *Liang et al.*, 1994; *Koster et al.*, 2000). These sub-grid schemes, however, were never designed to handle the sub-grid multi-scale coupling of the water, energy, and biogeochemical cycles as intended in contemporary applications (e.g., Earth System models, *Clark et al.*, 2015a). This is especially relevant as large scale models begin to include human impacts on land surface processes (*Wada et al.*, 2014; *Li et al.*, 2016). Acknowledging these constraints, in recent years there has been a renewed emphasis
on improving the representation of sub-grid heterogeneity through a more robust representation of soil, topographic, urban, and microclimate heterogeneity and by enabling explicit subsurface and surface interactions among sub-grid mosaic "tiles" or hydrologic response units (*Chaney et al.*, 2016a; *Subin et al.*, 2014; *Clark et al.*, 2015b).

Although these emerging approaches have the potential to considerably improve the representation of sub-grid heterogeneity in macroscale models, their added value depends on both the data and the approaches used to inform the sub-grid schemes
on the underlying heterogeneity of the physical environment—the primary driver of spatial heterogeneity in land surface processes (e.g., topography). The continual growth in the availability, detail, and wealth of Earth System data over the past decades provides an invaluable asset to make this possible. The use, harmonization, combination, and reinterpretation of field surveys, in-situ networks, and satellite remote sensing have led to petabytes of vegetation, topography, climate, meteorology, and soil data over continental to global extents with spatial resolutions ranging between 10-1000 meters (*Gesch et al.*, 2009;
*Chaney et al.*, 2016b; *Hengl et al.*, 2014; *Lehner et al.*, 2008; *Roy et al.*, 2010; *Pan et al.*, 2016; *Farr et al.*, 2007; *Fry et al.*, 2011; *Hijmans et al.*, 2005; *Chen et al.*, 2015; *Daly et al.*, 2008). These data, although uncertain, provide invaluable very high-resolution snapshots of the sub-grid physical environment and its impact on ecosystem spatial structure and function.

In most cases, environmental models use these data in two ways: 1) running the model at the native spatial resolution of the data and 2) running the model on a coarser grid by upscaling the data. Both options are inadequate for Earth system models;
the first is computationally unfeasible while the second mostly disregards the role of sub-grid heterogeneity. The question then remains: how can these data be used to the fullest extent while minimizing both computation and storage requirements?. This challenge is analogous to image compression where the goal is to maximize an image's information content while minimizing disk storage (e.g., clustering) (*Kanungo et al.*, 2002). For environmental data, this equates to effectively and efficiently summarizing the data while minimizing information loss. This concept is the underlying basis for mosaic schemes in land surface
models and hydrologic similarity in hydrologic models (*Avissar and Pielke*, 1989; *Beven and Kirby*, 1979).

Commonly, within macroscale models, the use of similarity amounts to binning (i.e., 1-dimensional clustering) maps of variables that are used as proxies of the drivers of spatial heterogeneity (e.g., topographic index is used to represent the role of topography on subsurface flow) to assemble a set of representative sub-grid tiles. However, recently there have been efforts to formally connect the concept of similarity to the clustering of an $n$-dimensional space—in this case, the $n$-dimensional space is
composed of the proxies of spatial heterogeneity (*Newman et al.*, 2014). *Chaney et al.* (2016a) takes this concept a step further



by building a model (HydroBlocks) that enables the explicit interaction among the tiles assembled via multivariate clustering—the connectivity between the tiles is learned from the elevation data. In this case, hydrologic connectivity was enforced in the clustering algorithm primarily through flow accumulation area derived from the DEM while mostly disregarding the basin's channel, hillslope, and sub-basin structure; this over-simplification of catchment structure can lead to overly complex and at times unrealistic inter-tile connections—critical to accurately simulating baseflow production. Thus the need remains for a clustering approach that allows for a minimal number of tiles while robustly accounting for the fine-scale hydrologic structure of the macroscale grid cell.

This paper introduces a hierarchical multivariate clustering approach (HMC) that summarizes the high dimensional environmental data space into hydrologically interconnected representative clusters (i.e., tiles). HMC has three main components: 1) cluster the fine-scale map hillslopes in a grid cell into characteristic hillslopes, 2) discretize the characteristic hillslopes into height bands, and 3) cluster the intra-band soil and vegetation. As a proof of concept, these clusters (i.e., tiles) are then used within the Geophysical Fluid Dynamics Laboratory (GFDL) LM4-HB land model to explore its potential to provide a robust multi-scale coupling between the water, energy, and biogeochemical cycles. Using a 1/4 degree (∼25 km) grid cell in southeastern California as a test-bed, this paper explores the number of tiles necessary to robustly account for the sub-grid multi-scale heterogeneity in the macroscale states and fluxes. It also explores the role that each of the drivers of spatial heterogeneity plays at the macroscale. Finally, the implications of this approach in the application and validation of large-scale environmental models and Earth System models are discussed.

## 2    Data

To develop, implement, and evaluate HMC, this study uses a 1/4 degree grid cell that covers the foothills and high sierras of southern Sierra Nevada in California (Figure 1). This domain is selected due to the observable role of the physical environment in the sub-grid heterogeneity. This heterogeneity is primarily explained by the strong topographic gradient between the Central Valley and the Sierra Nevada and its impact on precipitation and temperature; the highest point in the domain is 3118 meters while the lowest is 163 meters. This area has an annual average rainfall of 614 mm with large intra-cell variability with a minimum of 299 mm/year in the lowlands and a maximum of 1152 mm/year in the uplands. In both cases, most of the rainfall occurs between October and May. The uplands are primarily covered by evergreen vegetation while shrubs and grasses cover the lowlands. The uplands are characterized by a higher sand content than the lowlands and vice-versa for clay content.

### 2.1    Land and meteorological data

### 2.1.1    Topography

This study uses the 1 arcsec USGS National Elevation Dataset (NED) digital elevation model (DEM) and a series of derived products including flow accumulation area, hillslopes, slope, height above nearest drainage area (HAND), and aspect. The NED covers the contiguous United States (CONUS) and is created primarily from the USGS 10 m and 30 m digital elevation





models and from higher resolution data sources such as light detection and ranging, interferometric synthetic aperture radar, and high-resolution imagery (*Gesch et al.*, 2009).

## 2.2 Meteorology and climate

The recently developed Princeton CONUS Forcing (PCF) dataset provides a 1/32 degree (∼3 km) meteorological product over CONUS at an hourly temporal resolution between 2002 and present (*Pan et al.*, 2016). This dataset downscales the National Land Data Assimilation System phase 2 (NLDAS-2) product from 1/8 to 1/32 degree using a series of available products including Stage IV and Stage II radar/gauge products for rainfall. PCF includes precipitation, downward shortwave radiation, downward longwave radiation, air temperature, specific humidity, wind speed, and pressure. Furthermore, to inform the microclimate heterogeneity in HMC, this paper uses the recently released WorldClim 2 dataset. This gridded dataset derived from in-situ observations provides monthly climatologies of temperature, precipitation, solar radiation, vapor pressure, and wind speed over the global land surface at a 30 arcsec spatial resolution (*Fick and Hijmans*, 2017).

## 2.3 Soil properties

The soil properties come from the Probabilistic Remapping of SSURGO (POLARIS) dataset (*Chaney et al.*, 2016b), a new continental soil dataset that uses random forests to spatially disaggregate and harmonize the Soil Survey Geographic (SSURGO) database (*Soil Survey Staff*, 2013) over CONUS. In POLARIS, for each 30-meter grid cell, every soil series is assigned a probability of being found at a given grid cell. These probabilities are then combined with the vertical profile information available for each soil series to construct a minimum, maximum, weighted mean, and weighted variance for each grid cell—enough information to construct a beta distribution for each parameter per vertical layer of each grid cell. For this study, this approach provides porosity ($\theta_s$), wilting point ($\theta_{wp}$), and field capacity ($\theta_{fc}$), and saturated hydraulic conductivity ($K_{sat}$) which are set to be the median of each corresponding beta distribution. These data are then used to directly compute the bubbling pressure ($\psi_b$) and the inverse of the pore distribution size index ($B$) which are used in the Campbell water retention curve ($\theta = \theta_s(\psi_b/\psi)^{1/B}$) (*Campbell*, 1974).

### 2.3.1 Land cover

The Cropland Data Layer (CDL) provides the 30-meter land cover types. The CDL is an annually-produced database over CONUS that combines the National Land Cover Database (*Fry et al.*, 2011) with an annual analysis of the spatial distribution of croplands. It is created and managed by the United States Department of Agriculture's National Agricultural Statistic Service (USDA-NASS). The predicted land cover types are based on the reflective signatures from a number of satellites including Landsat TM and ETM+, MODIS satellite data, and the Advanced Wide Field Sensors (AWiFS), among others (*Boryan et al.*, 2011). The different categories are associated to their corresponding land use types and species within the LM4-HB model.



## 3 Methods

### 3.1 Land model description: LM4-HB

For this study, the conceptual approach that is used to parameterize sub-grid heterogeneity in the HydroBlocks land surface model (HB, *Chaney et al.*, 2016a) is added to the fourth generation of the Geophysical Fluid Dynamics Laboratory (GFDL) land model (LM4, *Shevliakova et al.*, 2009; *Milly et al.*, 2014; *Subin et al.*, 2014). The resulting LM4-HB model is used to explore how Big Data can be efficiently and effectively harnessed to improve the characterization of the sub-grid multi-scale heterogeneity in Earth System models. LM4-HB uses a hierarchical approach to represent the underlying sub-grid heterogeneity; this makes it an ideal candidate for the testing and implementation of a hierarchical multivariate clustering algorithm to assemble the underlying sub-grid heterogeneity from available environmental datasets. This section provides an overview of LM4-HB with a primary focus on describing its hierarchical representation of sub-grid heterogeneity.

The land fraction of each grid cell in LM4-HB is partitioned into soil, glacier, and lake components. The soil component in turn is composed of $k$ characteristic hillslopes; each hillslope has a unique set of attributes including slope, aspect, convergence, and convexity, among others. As shown in Figure 2, each characteristic hillslope $i$ is divided into $l_i$ height above nearest drainage area (HAND) bands (referred to from now on as height bands). Each height band $b_{i,j}$ is divided into $p_{i,j}$ clusters (i.e., tiles) to account for the intra-band heterogeneity in soil and land cover. The total number of soil tiles is given by the sum of all tiles in each height band for all characteristic hillslopes within a grid cell. Although not shown in Figure 2, the current model uses a uniform soil depth for all tiles within a characteristic hillslope. However, the soil properties (e.g., porosity and hydraulic conductivity) control the effective soil depth of each tile and thus variable soil depths as shown in the schematic are effectively represented.

Each soil tile within LM4-HB consists of a model from canopy air down to impermeable bedrock. The processes captured within the model include bidirectional diffuse and direct, visible and near-infrared radiation transfer; photosynthesis and stomatal conductance; surface energy, momentum, and water fluxes; snow physics; soil thermal and hydraulic physics (including advection of heat by water fluxes); vegetation phenology, growth, and mortality; simple plant-functional-type transition dynamics; and simple soil-carbon dynamics. For further details on the intra-tile processes see (*Shevliakova et al.*, 2009; *Milly et al.*, 2014).

Within each characteristic hillslope, each tile interacts with the tiles in its same height band and the tiles in the height bands immediately below and above via the subsurface flow of water; heat and carbon are advected by the water fluxes. The tiles adjacent to the channel interact with the stream in one direction (tile to stream). Each height band is characterized by a length, width, and height above nearest drainage area. The effective width of a tile for a given height band is the corresponding fraction of the width of the height band. For further details on the tile interactions and the hillslope model more generally see *Subin et al.* (2014).

For all model simulations in this study, LM4-HB is run with a 50 meter soil depth (the same for all tiles) at a 1 hour time step for 130 years by cycling through the forcing between 2002 and 2014 ten times. The first 117 years are used for spin-up while the final 13 years of the simulation are used for the analysis.





## 3.2 Assembling the land model tiles: Hierarchical multivariate clustering

To take advantage of LM4-HB's sub-grid representation of land cover, soil, climate, and topography, the characteristic hillslopes and the intra-hillslope heterogeneity are parameterized using available continental and global environmental data. This section provides an overview of the hierarchical multivariate clustering (HMC) algorithm used to assemble a grid cell's tiles.

Its steps are: 1) define the characteristic hillslopes, 2) discretize the characteristic hillslopes into height bands, and 3) define the intra-band heterogeneity.

### 3.2.1 Define the characteristic hillslopes

The characteristic hillslopes for a given grid cell are defined by clustering a grid cell's fine-scale map of hillslopes. To assemble the map of hillslopes, the DEM is sink-filled (*Planchon and Darboux*, 2002) and the channels are delineated using an area

threshold of 100,000 m$^2$. A recursive algorithm then splits each basin into a maximum of three hillslopes—left side, right side, and headwaters. Each hillslope's attributes are assembled from the high-resolution soil, topography, and climate data. These include each hillslope's average aspect, slope, annual mean precipitation, and annual mean temperature. Metrics that summarize each hillslope's plan and profile geometry are derived from the hillslope's binned HAND data. Given each bin's slope, HAND, and area, the length and width are readily computed. To summarize these properties, a line is fit to the set of widths for each

hillslope; the slope of this function provides a summary metric of the hillslope's plan shape (convergence/divergence). For each hillslope's profile, the function $h = H\left[1 - (1 - (x/L)^a)^b\right]$ is fit to the binned HAND data; where $h$ is the HAND, $H$ is the maximum HAND, $x$ is the horizontal position, and $L$ is the hillslope length. The parameters $a$ and $b$ summarize the concavity of the lower half and upper half of each hillslope respectively. This function is chosen due to its flexibility to reproduce convex, concave, and complex hillslope profiles.

Assembling all the calculated attributes leads to an $n$ by $m$ array where $n$ is the number of hillslopes and $m$ is the number of attributes. The k-means clustering algorithm (*MacQueen*, 1967) is then used to partition the normalized $m$ dimensional attribute space into $k$ characteristic hillslopes. Figure 3 provides an overview of the steps used to define the characteristic hillslopes. The attributes of each characteristic hillslope are set to be the arithmetic mean of the attributes of its corresponding hillslopes.

### 3.2.2 Discretize the characteristic hillslopes

After assembling the set of characteristic hillslopes, their attributes, and their corresponding profile and width functions, the next step is to discretize each profile along the length axis into height bands. The number of height bands is $l_i = \lceil H_i/\Delta h \rceil$; where $i$ is the characteristic hillslope, $H_i$ is the profile's maximum height, and $\Delta h$ is the height difference between adjacent height bands. Note that the number of height bands per characteristic hillslope can differ per characteristic hillslope since each

one has a unique $H_i$.

Using the high-resolution maps of characteristic hillslopes and the HAND, each high-resolution grid cell is assigned a characteristic hillslope and a height band. This connects the discretized characteristic hillslopes to the observed landscape.



Figure 4 illustrates an example discretization of a characteristic hillslope and the mapping of the discretized hillslopes to the high-resolution grid.

### 3.2.3 Define the intra-band heterogeneity

The final step is to define the heterogeneity within each height band $b_{i,j}$ where $i$ is the characteristic hillslope and $j$ is the height
band. For each band $b_{i,j}$, the collocated fine-scale grid cell values of the proxies of heterogeneity are extracted. This study uses saturated hydraulic conductivity, porosity, and a set of binary maps (natural/cropland, evergreen/deciduous, and grass/tree) derived from the high resolution land cover map. For each height band $b_{i,j}$, this leads to an $n$ by $m$ array of attributes where $n$ is the number of fine-scale grid cells that belong to $b_{i,j}$ and $m$ is the number of attributes. The k-means algorithm is then used to cluster the $m$ dimensional attribute space into $p_{i,j}$ intra-band clusters (i.e., tiles). In this study, for simplicity, $p_{i,j}$ is set to be
the same for all height bands and characteristic hillslopes. Therefore, the final number of tiles for the macroscale grid cell is given by:

$$n_{tiles} = p \sum_{i=1}^{k} l_i \tag{1}$$

Each tile within a grid cell is assigned an id $t_{i,j,k}$ where $i$ is the characteristic hillslope, $j$ is the height band, and $k$ is the intra-band cluster. Figure 5 shows an example of the intra-band clustering and the tile configuration resulting from the hierarchical
clustering algorithm over this study's domain. For each tile, the land model parameters are set to be the arithmetic average of all the parameter values of the fine-scale grid cells that belong to the given tile. Each tile is assigned its own meteorology by assigning a weighted average of all the overlying 4 km PCF grid cells that intersect with the 30-meter grid cells that belong to the tile. One of the advantages of having each 30-meter grid cell belong to a tile and the corresponding map of tiles is that the simulations for each tile can be mapped out to the fine scale grid to provide a 30-meter representation of the model output at
each time step.

### 3.3 Model experiments

### 3.3.1 Exploratory simulation

By clustering the high-dimensional environmental space, HMC explicitly relates the tiles used in LM4-HB to the observed fine-scale physical environment while ensuring realistic hydrologic connections between tiles along characteristic hillslopes.
To illustrate the benefits and additional model information that can be extracted when using HMC, an exploratory simulation is run using a simple HMC-assembled tile configuration ($k = 2$, $\Delta h = 50$ meters, $p = 2$) with 14 tiles within LM4-HB. This tile configuration is among the simplest case for this domain that is able to illustrate the role of all the different drivers of heterogeneity. This exploratory simulation is then analyzed to illustrate the added information that HMC provides.



### 3.3.2 Hierarchical multivariate clustering: Parameter sensitivity

The primary objective of HMC is to harness high-resolution environmental data to efficiently and effectively summarize a macroscale grid cell's underlying multi-scale spatial structure. To make this possible, HMC relies on a set of user-defined parameters to control the importance of each hierarchical step in the algorithm; these parameters include: 1) the number of

characteristic hillslopes ($k$), 2) the elevation difference between adjacent height bands ($\Delta h$), and 3) the number of intra-band clusters per height band ($p$). To test LM4-HB's sensitivity to these parameters, 9 model experiments are performed in which the HMC parameters are adjusted to assess their individual roles. The model experiments are outlined in Table 1. Experiments $e_{1,1000,1}$, $e_{2,1000,1}$, and $e_{10,1000,1}$ increase $k$ from 1 to 10 characteristic hillslopes, experiments $e_{10,50,1}$, $e_{10,20,1}$, and $e_{10,10,1}$ decrease $\Delta h$ until 10 meters, and experiments $e_{10,10,2}$, $e_{10,10,3}$, and $e_{10,10,5}$ increase $p$ up to 5 intra-band clusters.

### 3.3.3 Characterizing the roles of the drivers of heterogeneity

Applying HMC on existing high resolution environmental data enables a robust representation of the different drivers of spatial heterogeneity (soil, topography, meteorology, and land cover) within macroscale environmental models. However, it does not explicitly characterize the individual role of each driver—key to advancing our understanding of the relationship between the physical environment, ecosystem spatial structure, and macroscale response. To make this analysis possible, another set of

model experiments are explored that investigate the individual role of each driver at the macroscale. Using an approach similar to *Chaney et al.* (2014), each driver's sensitivity is explored by turning the heterogeneity of properties associated with each driver on and off. When "on" the properties associated to the driver are left as assigned through HMC; when "off" the driver's properties are set to be the grid cell mean. The different model experiments are outlined in Table 2.

## 4 Results and Analysis

### 4.1 Exploratory simulation

Figure 6 shows the simulated time series for all 14 tiles at a daily time step for baseflow, root zone soil moisture, evapotranspiration, and sensible heat flux. The fine-scale map of tiles is also shown and color-coded to relate the spatial location of each tile to the simulated time series. Each tile is assigned an id $t_{i,j,k}$, where $i$ is the characteristic hillslope, $j$ is the height band, and $k$ is the intra-band cluster. A comparison of the tiles' time series exemplifies the differences in states and fluxes that are driven

by a tile's location, properties, and meteorology. Explicitly resolving hillslope dynamics leads to significant differences in the root zone soil moisture; in general, soil moisture decreases, as the tiles are further away from the valley. However, land cover, soil, hillslope structure, and meteorological differences can lead to differences in soil moisture between hillslopes and intra-band clusters (e.g., uplands vs lowlands). This strong topographic gradient in soil moisture provides more water for vegetation growth in the valleys than along the ridges; given that this is a water-limited area, this explains the appreciable differences in simulated evapotranspiration. Furthermore, this also explains the strong heterogeneity in sensible heat flux with tiles' with





higher soil moisture having lower sensible heat fluxes. Finally, as expected, only the tiles that are adjacent to the channels ($t_{1,1,1}$, $t_{1,1,2}$, $t_{2,1,1}$, and $t_{2,1,2}$) have a baseflow signal.

For all states and fluxes, the macroscale (tile weighted average) time series for each variable is superimposed on the tile simulations to illustrate the strong differences that can exist between individual tiles and the macroscale estimate (temporal

dynamics and mean). These differences illustrate the challenge of comparing a macroscale estimate to observations and simulations at different spatial resolutions; a persistent challenge when aiming to apply and evaluate macroscale models. However, as will be discussed in section 5.2, being to able to connect each 30-meter grid cell to each tile simulation enables a path towards circumventing the scale discrepancies between macroscale model estimates and in-situ observations.

Figure 7 illustrates how the tile simulations can then be mapped out onto the 30-meter fully distributed grid. In this example,

the daily averaged simulated evapotranspiration value for each tile on June 16th, 2005 is assigned to each corresponding fine-scale grid cell. Being able to visualize the assumed heterogeneity of the model enables a more realistic comparison to fully distributed models. Furthermore, it makes it possible to provide model output at spatial resolutions at far finer spatial resolutions than the tile weighted average (i.e., macroscale estimate).

### 4.2 Hierarchical multivariate clustering: Parameter sensitivity

As outlined in Section 3.3.2, the experiments in this section explore the sensitivity of the HMC parameters through a set of model experiments; these experiments are summarized in Table 1.

As an initial visual comparison, Figure 8 shows the mapped out annual mean evapotranspiration between 2002 and 2014 at a 30-meter spatial resolution for the different model experiments. The baseline experiment is the one tile configuration (i.e. no sub-grid heterogeneity). An increase in the number of characteristic hillslopes leads to the appearance of large-scale

spatial patterns in evapotranspiration (experiments $e_{1,1000,1}$, $e_{2,1000,1}$, and $e_{10,1000,1}$) . This is primarily due to the strong topographic gradient in precipitation between the lowlands and uplands; this heterogeneity in evapotranspiration is possible through the disaggregation of the PCF meteorology among the land model tiles. Decreasing $\Delta h$ leads to an increase in the number of height bands per characteristic hillslope; this makes the role of topographic convergence in sub-surface flow readily apparent—evapotranspiration is higher in the riparian zones (experiments $e_{10,50,1}$, $e_{10,20,1}$, and $e_{10,10,1}$). Finally, increasing the

number of intra-band clusters adds to heterogeneity in evapotranspiration due to land cover and soil heterogeneity (experiments $e_{10,10,2}$, $e_{10,10,3}$, and $e_{10,10,5}$). Note how after experiment $e_{2,1000,1}$ (278 tiles), the spatial patterns cease to change as much.

Figure 9 formalizes the comparison between the different model experiments; it shows how the spatial mean and spatial standard deviation of the annual mean of each year between 2002 and 2014 change as a function of the tile configuration for a suite of states and fluxes. The primary result is the apparent convergence of all states and fluxes for both the spatial mean and

standard deviation with the increase in the number of land model tiles. In other words, there is a point at which further increases in the number of tiles have a limited impact at the macroscale. For this site, that limit is approximately 300 tiles compared to the 810,000 fine-scale grid cells in the domain. This result is encouraging; it illustrates that multi-scale sub-grid heterogeneity can be characterized effectively and efficiently in large scale models by taking advantage of the covariance between environmental properties.





The role that each parameter of HMC has at the macroscale depends on the prognostic variable; these differing roles are discussed below.

- $k$ - An increase in the number of characteristic hillslopes from experiments $e_{1,1000,1}$ (1 tile) to $e_{10,1000,1}$ (10 tiles) leads to noticeable changes in all the prognostic variables. The most noticeable changes occur when increasing the number of characteristic hillslopes from 1 to 2. This is primarily due to the improved representation of land cover heterogeneity; instead of the grid cell being represented uniformly as evergreen trees, the lowlands are now grasses while the uplands remain as evergreen trees. This leads to a decrease in the cell's effective roughness length (i.e., a decrease in aerodynamic conductance), and thus a decrease in sensible heat flux and an increase in surface temperature. The decrease in aerodynamic conductance also contributes to a decrease in transpiration.

- $\Delta h$ - A decrease in $\Delta h$ from experiments $e_{10,50,1}$ (28 tiles) to $e_{10,10,1}$ (139 tiles) leads to an increase in the number of height bands in the characteristic hillslopes. This results in an explicit representation of the role of topographic convergence in sub-surface flow; more soil water is available in the valleys than along the ridges. The increase of soil water in the valleys also leads to more frequent saturated excess runoff; to the extent that during wet years this counteracts the decrease in baseflow. Another noticeable impact of the increase in the number of height bands is the decrease in inter-annual variability in transpiration, net primary productivity, baseflow, and sensible heat flux. As will be discussed in Section 5.1 this can have potentially important implications for the role of ecosystem spatial structure on ecosystem resilience to hydrologic extremes.

- $p$ - An increase in the number of intra-band clusters from experiments $e_{10,10,2}$ (278 tiles) to $e_{10,10,5}$ (695 tiles) leads to a more robust representation of soil and land cover heterogeneity throughout the domain. This leads to differences in most variables. However, these differences are not as noticeable as those due to changes in $k$ and $\Delta h$ since most of the heterogeneity in land cover and soil heterogeneity has already been represented through these other parameters. This parameter will most likely play a larger role in regions where the ecosystem spatial structure is not as strongly tied to topography.

### 4.3 Characterizing the roles of the drivers of heterogeneity

Although Section 4.2 provides preliminary insight into the role of the different drivers of spatial heterogeneity (e.g., topographic convergence impacts the macroscale soil moisture mean), due to the interactions of the HMC parameters, it cannot precisely disentangle each driver's unique role. For this purpose, as introduced in Section 3.3.3, another set of experiments are explored that turn the different drivers of heterogeneity on and off. These experiments are summarized in Table 2. Furthermore, the tile configuration of experiment $e_{10,10,2}$ in Section 4.2 (278 tiles) is used for all experiments in this section; this tile configuration is chosen because, as shown in Section 4.2, the model macroscale states and fluxes converge at around 300 tiles.

As an initial visual comparison, the mapped out model results of the different experiments are shown in Figure 10 for annual mean runoff, 2 meter root zone soil moisture, leaf area index, and soil temperature between 2002 and 2014 at a 30-meter spatial



resolution; Figure 11 formalizes this comparison. The role that each driver plays on the macroscale prognostic states and fluxes are discussed below.

– **Baseline (B)** - The baseline experiment equates to the homogeneous sub-grid cell case. However, in this case, there are 278 tiles where each tile has the same soil properties, hillslope structure (each tile is set to be its own characteristic hillslope), and land cover properties. Furthermore, each tile is run using the grid cell mean meteorology. Not surprisingly, there is no heterogeneity in the plotted maps and the spatial standard deviation for all prognostic states and fluxes is 0.

– **Soil heterogeneity (S)** - This experiment adds heterogeneity in the soil properties; this includes porosity, and the Campbell retention curve parameters, among others. It has a relatively small impact on the spatial mean for all variables. However, there are changes in their spatial standard deviations. The increase in spatial standard deviation is largest for soil moisture, which in turn impacts the remaining prognostic variables. For example, changes in available water impacts infiltration excess runoff leading to, at times, appreciable heterogeneity in annual runoff production. In any case, these differences are minor at the macroscale when compared to the subsequent experiments.

– **Soil and hillslope heterogeneity (SH)** - Assigning each tile to its original corresponding characteristic hillslope and discretizing these hillslopes leads to significant differences at the macroscale. The strong topographic gradients caused by the discretized hillslopes lead to strong topographic gradients in soil moisture and thus explain the sharp increase in the spatial standard deviation of soil moisture. These topographic gradients in soil moisture lead to an overall increase in saturated excess runoff during wet years, thus counteracting the overall decrease in baseflow; this also explains the increase in the spatial standard deviation of runoff. The most noticeable role of the topographically driven subsurface flow is the reduction in inter-annual variability in root zone soil moisture, leaf area index, baseflow, transpiration, net primary productivity, and sensible heat flux. This is due primarily to the significant increase in inter-annual change in storage when including explicit topographic gradients; these topographic gradients enable the system to be able to release more water during dry years (uphill deep soil water is made available to the riparian zone through subsurface flow) and to absorb more water during the wet years (increase in infiltration capacity due to the heterogeneity of soil moisture).

– **Soil, hillslope, and land cover heterogeneity (SHL)** - This experiment adds heterogeneity in land cover. These changes are similar to those seen in Section 4.2 when increasing the number of characteristic hillslopes. This is because both cases ensure evergreen forests are represented in the uplands and grasslands and shrubs are represented in the lowlands. This leads to a lower effective roughness height explaining the decrease in sensible heat flux, net primary productivity, and transpiration. Land cover heterogeneity also leads to an appreciable increase in the spatial standard deviation of sensible heat flux, net primary productivity, and transpiration. Its role is also particularly noticeable in the soil temperature spatial distribution by creating sharp contrasts between the lowlands and uplands.

– **Soil, hillslope, land cover, and meteorological heterogeneity (SHLM)** - This last experiment adds heterogeneity in meteorology by prescribing the meteorology to each tile from the overlying 4 km PCF grid. This leads to a net increase in annual mean runoff and a decrease in transpiration. The spatial heterogeneity of meteorology further enhances the



contrast in soil temperature between the lowlands and uplands. Furthermore, the larger availability of water in the uplands leads to higher net primary productivity in this region and thus higher LAI.

## 5   Discussion

### 5.1   Sub-grid redistribution of water: Dampening of extremes

Seeking to account for the role of sub-grid redistribution of subsurface water within macroscale hydrologic and land surface models is not a new objective; many schemes have been implemented over the past decades to characterize its influence (*Liang et al.*, 1994; *Beven and Kirby*, 1979; *Milly et al.*, 2014). However, these approaches are designed primarily to account for the role of fine-scale heterogeneity on hydrologic response; they are not meant to handle the sub-grid spatial coupling of the water, energy, and biogeochemical cycles. LM4-HB addresses these limitations by explicitly modeling the subsurface flow of water
via horizontally and vertically discretized characteristic hillslopes (*Subin et al.*, 2014); this makes it possible to account for the impact of sub-grid redistribution of water on the full gamut of land surface states and fluxes (e.g., soil moisture, sensible heat flux, and net primary productivity). Furthermore, by harnessing existing environmental information to parameterize sub-grid heterogeneity, HMC ensures that the properties of the characteristic hillslopes are formally connected to the observed physical environment.

The results from the model experiments in Section 4.3 show that upon enabling subsurface redistribution, the most noticeable difference at the macroscale is the dampening of annual extremes in the water, energy, and carbon cycles; for example, as shown in Figure 11 there is a strong decrease in the inter-annual variability of baseflow, transpiration, net primary productivity, and sensible heat flux between the S and SH simulations. As mentioned in Section 4.3, this is primarily due to the significant increase in inter-annual change in storage when including explicit topographic gradients; these topographic gradients enable the
system to be able to release more water during dry years (uphill deep soil water is made available to the riparian zone through subsurface flow) and to absorb more water during the wet years (increase in infiltration capacity due to the heterogeneity of soil moisture). Although this role of topography in infiltration capacity and baseflow production has been recognized for decades in hydrology (*Liang et al.*, 1994; *Beven and Kirby*, 1979; *Koster et al.*, 2000), to the authors knowledge, this study is the first to explicitly explore its role in the coupled water, energy, and biogeochemical cycles.

These model experiments provide insight into the role of subsurface redistribution at the macroscale; the underlying physical environment provides a mechanism for ecosystems with pronounced topography to mitigate the impacts of seasonal to annual hydrologic extremes. These results suggest that an improved representation of spatial heterogeneity could improve projections of ecosystem response to drought, particularly in mountainous regions. A better representation of biophysical feedbacks to variations in air temperature and vapor pressure deficit could also improve simulations of land-atmosphere feedbacks that
can intensify droughts and affect macroscale circulations (*Bagley et al.*, 2012; *Nicholson*, 2015; *Berg et al.*, 2015). Future work should explore how these results extend to other regions with different configurations of the physical environment (e.g., topography). Beyond understanding the role of the sub-grid subsurface redistribution of water, these model experiments would also bring to light other impacts that the fine-scale physical environment has on macroscale response.





## 5.2 Revisiting the application and evaluation of Earth System models

As explored in Section 4.1 and shown in Figure 7, formally connecting the sub-grid tile configuration to the high-resolution environmental covariates provides a novel approach to visualize the output of the land components of Earth System models—the simulation of each land model tile can be mapped to the fine-scale grid. Using this approach, macroscale models are able

to maintain their existing computational and storage efficiency while providing highly detailed local information. As discussed below, this has important repercussions for the evaluation and application of these models.

**Evaluation** - As shown in Figure 6, when robustly characterized, sub-grid multi-scale heterogeneity can lead to significant differences between the simulations at the tile (i.e., field-scale) and grid cell levels (i.e., macroscale). This discrepancy in spatial scale is analogous to using in-situ observations to evaluate and validate Earth system models: the observations are at the

field-scale yet the model estimates are at the macroscale. The approach explored in this study allows the Earth system modeling community to revisit this persistent challenge. Since each fine-scale grid cell ($\sim$30 meters) is assigned to a land model tile, in-situ observations can be readily compared to the simulations of their collocated model tile. This makes it possible to evaluate these models using in-situ observation networks (e.g., FLUXNET) without having to upscale the observations or downscale the macroscale estimate. Furthermore, it also enables the use of very high-resolution satellite products (e.g., Landsat) to evaluate

the modeled fine-scale spatial patterns. This approach enables the Earth system modeling community to work more closely with field scientists to further understanding of the Earth System and to accelerate the model development cycle.

For example, this approach could facilitate improved methods for validating soil carbon projections in ESMs. Past model-data comparisons of soil carbon have relied on spatial models to scale soil carbon measurements to the grid cell scale, as in the Harmonized World Soil Database (HWSD; *Todd-Brown et al.* (2013); *Luo et al.* (2016)). Such scaling techniques can be

problematic for direct comparison with model simulations. First, the spatial models necessary for scaling have the potential to introduce bias in the observation-based product, and result in a comparison that is not purely measurement-based. Second, variability in topography, ecosystem type, and soil properties within the grid cell scale makes these comparisons challenging to interpret: failure of a model grid cell to match scaled observations could be due to process representation, model parameterization, or the relative spatial coverage of ecosystem or edaphic types within the model grid cell relative to the scaled observations.

Incorrect attribution of model error could result in inappropriate adjustments to model parameters or processes, for example reducing the turnover rate of soil carbon pools in uplands when a model underestimate of carbon stocks is actually due to high carbon stocks observed in wetlands. HMC could address these issues by facilitating direct comparison of modeled soil carbon stocks with measurements grouped by the same properties used in the clustering analysis.

**Application** - Earth System models are used almost exclusively for regional to global applications due to their coarse spatial

scales with limited use by local stakeholders (e.g., farmers). Having the ability to map the tile simulations to the fine-scale grid ($\sim$30 meters) allows the community to reevaluate how these models are applied. For example, accounting for the very high resolution soil properties in each model tile leads to more locally-relevant soil moisture simulations; providing these field-scale model estimates in real-time can then be used to inform irrigation requirements. Furthermore, the inherent model efficiency of this approach facilitates robust ensemble frameworks; this enables a path towards constraining the unavoidable



uncertainty of the model predictions—even more pervasive at higher spatial resolutions—while still providing local detail. This novel approach to model application should be explored further as it requires minimal increases in computational expense with potentially large societal benefits.

## 5.3 Hierarchical multivariate clustering: Parameter optimization

Section 4.2 illustrates how a relatively small number of land model tiles are necessary to explicitly characterize the underlying sub-grid heterogeneity in this study's domain—this is substantially less than the 810,000 grid cells of a corresponding 30 meter fully distributed simulation. This provides the best trade-off for large-scale models: it explicitly captures the role of sub-grid fine-scale features while mantaning computational efficiency. However, although this study does provide a preliminary exploration of the HMC parameter space ($k$, $\Delta h$, and $p$), it does not provide a robust approach to find the optimal HMC parameters at different regions over the globe. Three approaches that are currently being explored to accomplish this goal are outlined below.

1. The most direct approach is to optimize the HMC parameters on all macroscale grid cells for a given grid size over the globe. This would be accomplished through parameter optimization techniques (e.g., *Hadka and Reed*, 2013; *Duan et al.*, 1993). For each parameter set, HMC would be run to create the model tiles and then LM4-HB would be used to run a simulation to characterize its long-term macroscale states and fluxes. Convergence on the fully distributed simulation would be attained at the parameter set that leads to the fewest number of tiles while ensuring the macroscale states and fluxes have converged within a user-defined tolerance. This would provide a robust solution albeit requiring substantial computational resources.

2. A more computationally efficient path forward is to use the first approach on only a subset of macroscale grid cells or catchments. These domains would be chosen such that they sample comprehensively from different climate, soil, land cover, and topographic regimes throughout the globe. The HMC parameters would be optimized independently for each of these domains. Machine learning (e.g., random forests) would then be used to regionalize these optimized parameters by deriving non-linear functional relationships between the optimized parameters and a suite of summary macroscale metrics (e.g., standard deviation of elevation, grid cell area, and average precipitation, among others). This approach would provide a method to assemble the optimal HMC parameters for a chosen region without having to resort to optimizing the parameters for a new domain.

3. Finally, another option is to rely exclusively on the data that is used within HMC. The primary goal behind clustering these data is to extract all the relevant information and minimize redundancies. Assuming that the water, energy, and biogeochemical fine-scale features are tightly coupled to the observed environment, then ensuring that the mapped out clustered input data matches the original fine-scale maps would ensure that the model results will provide a robust approximation of the fully distributed model. This approach would provide a method to define the optimal number of land model tiles without having to resort to model simulations; thus making the parameter selection less model dependent.



## 5.4 Hierarchical multivariate clustering: Expanding beyond natural soil systems

The strong covariance between the different proxies of the drivers of spatial heterogeneity makes it possible to summarize a high dimensional proxy space (i.e., environmental data) using a relatively small number of representative clusters. HMC capitalizes on this covariance to characterize the fine-scale heterogeneity of natural soil systems. However, the covariance of environmental properties is not exclusive to natural soil systems and can be extended to other systems over land including urban areas, croplands, water bodies, and glaciers. This section explores both the data that is available for these types of systems and how clustering can be used to extract their most representative characteristics.

- **Lakes and glaciers** - Earth System models represent lakes and glaciers as model tiles over land. Each lake tile is characterized by a set of properties including depth and area; these can be obtained from existing global lake inventory databases. These data have information such as shoreline length, water volume, and average depth, among others (e.g., HydroLakes; *Messager et al.* (2016)). For each macroscale grid cell, clustering could be used to define a set of characteristic lakes with their associated representative properties. A similar approach could also be used to identify a grid cell's characteristic glaciers by clustering the properties associated associated with the glaciers within that cell; rich global glacier inventory databases such as GLIMS (*Raup et al.*, 2007) could then be harnessed within ESMs.

- **Urban areas** - Recognizing the important role that urban areas play in the coupled system, the community is actively incorporating them into ESMs through urban canopy models (UCMs) (*Li et al.*, 2016). UCMs represent urban areas through a set of characteristics including roof fraction, building height, and canyon fraction, among others. Although there is currently no comprehensive database that provides the characteristics for all urban areas over the globe, there are emerging efforts to make this information available (e.g., WUDAPT; *Bechtel et al.* (2015)). As this data becomes accessible, clustering could also be used to distinguish the characteristic urban features within a model grid cell (e.g., high vs. low buildings).

- **Croplands** - Although Earth System models include croplands, their representation is generally oversimplistic. For example, in most cases, irrigation practices are ignored and many different phenological and physiological differences between crops are disregarded (e.g., rice vs. corn). Although far from complete, datasets are emerging that are able to provide this information at moderate to very high spatial resolutions over continental to global extents (*Boryan et al.*, 2011; *Pervez and Brown*, 2010; *Siebert et al.*, 2013; *Teluguntla et al.*, 2015). These data provide metrics that summarize local cropland characteristics (e.g., C3/C4) and irrigation practices (e.g., irrigated/rainfed). This information can be summarized per grid cell via clustering to provide a more complete representation of sub-grid cropland characteristics. Although this study explores this possibility using the CDL database, further work is necessary to more adequately account for the sub-grid variablity in crop characteristics and irrigation practices.

- **Peatland and permafrost landscapes** - In wet and high-latitude regions characterized by organic matter accumulation in peat and permafrost, spatial heterogeneity can be crucial to understanding regional carbon stocks. For example, *Buffam et al.* (2011) found that peat and lake sediments account for more than 70% of carbon stocks despite covering only 33% of





the land area in a northern Wisconsin landscape. Likewise, in permafrost systems microtopographic variations driven by ice-wedge polygon formation can dominate spatial variability of carbon cycling (*Lipson et al.*, 2012; *Zona et al.*, 2011). While spatially-explicit modeling of these complex landscapes can yield important insights about ecosystem function and vulnerability to climatic changes (e.g., *Sulman et al.*, 2012; *Sonnentag et al.*, 2008), the reduced computational demands of HMC could facilitate incorporating these important dynamics into macroscale simulations.

## 6 Conclusions

A robust representation of the influence of the multi-scale physical environment on the coupled terrestrial water, energy, and biogeochemical cycles remains a persistent challenge in Earth System models. One of the principal obstacles is the oversimplification of the observed complex heterogeneity within these models. This is primarily due to a limited understanding of how to use the available petabytes of environmental data effectively and efficiently in macroscale models.

Unsupervised machine learning, and more specifically cluster analysis, provides a path forward by capitalizing on the observed landscape similarity to extract the underlying defining features (i.e., clusters) from available environmental data. The hierarchical multivariate clustering (HMC) approach presented here takes this a step further by taking advantage of these clustering techniques while also ensuring physically consistent surface and subsurface interactions between clusters through discretized characteristic hillslopes.

A series of different tile configurations computed via HMC are used within the LM4-HB model to quantify its added-benefits to macroscale models. The model experiments over a 1/4 degree in southeastern California show that: 1) the observed similarity over the landscape makes it possible to robustly account for the role of multi-scale heterogeneity in the macroscale states and fluxes with a relatively minimal number of sub-grid land model tiles; 2) assembling the sub-grid tiles from the observed high-dimensional environmental data can lead to important differences in the macroscale water, energy, and carbon cycles; 3) connecting the fine-scale grid to the model tiles via HMC makes it possible to circumvent the scale discrepancies between the macroscale and field-scale estimates—this has significant implications for how Earth Systems models are evaluated and applied.

HMC illustrates a path towards improving the representation of land heterogeneity in ESMs by harnessing the available petabytes environmental data. However, HMC only scratches the surface of what is possible. Moving forward, these approaches could be extended beyond natural systems to managed systems (urban areas, croplands, reservoirs, and pumping, among others). Clustering techniques could also be applied for non-soil systems including lakes and glaciers. Furthermore, although using hillslopes as the governing hydrologic structures is appropriate for the domain used in this study, this will not be true everywhere (e.g., flat terrain). In these cases, the hierarchical approach could be relaxed or extended to include other structures including stream orders and basins.

Finally, the volume and complexity of data available today pales in comparison to what will be available in the coming decades (*McCabe et al.*, 2017). These data will provide unique opportunities for Earth System science; however, unless methods are developed that can harness these data, their intrinsic value to improve our understanding of the Earth System will be limited.



We encourage the Earth system modeling community to pursue the use of clustering techniques to ensure these data are used effectively and efficiently in macroscale models.



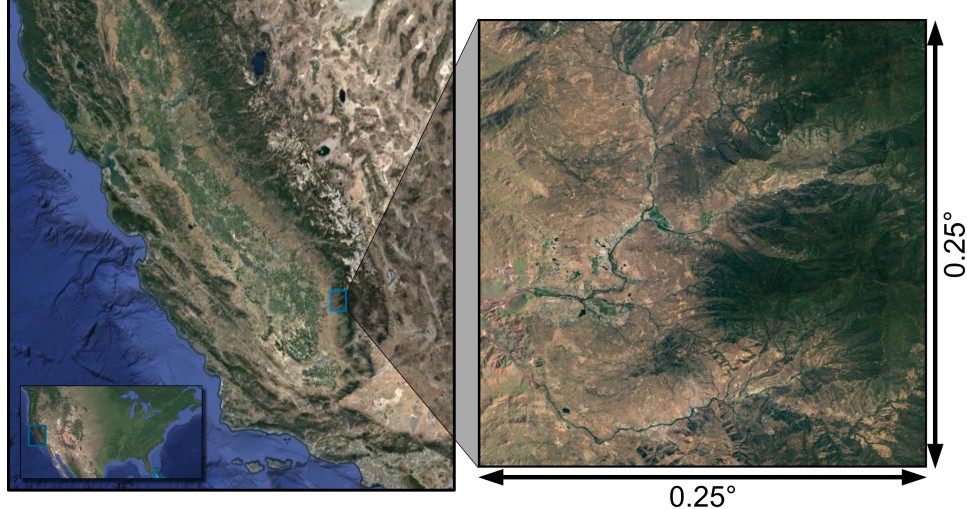

**Figure 1.** Test-bed site in the foothills of southern Sierra Nevada in California used to develop, implement, and test the HMC algorithm. The region is characterized by strong heterogeneity in topography, climate, and soil properties leading to a complex multi-scale ecosystem spatial structure.



# Characteristic hillslope

**Figure 2.** Schematic representation of a characteristic hillslope. Each characteristic hillslope is divided into height bands; which in turn are partitioned into intra-band tiles. Each tile interacts via the subsurface flow of water with the tiles in its same height band and with all the tiles in the height bands below and above.





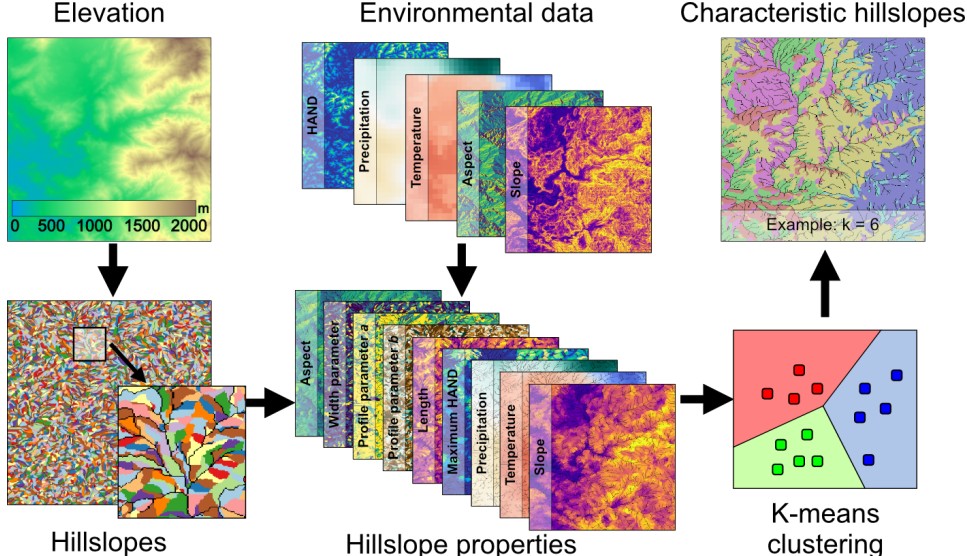

**Figure 3.** The characteristic hillslopes are defined by: 1) delineating the hillslopes from the elevation data, 2) calculating a suite of properties for each hillslope from environmental data, and 3) using the hillslope properties and the k-means clustering algorithm to define $k$ characteristic hillslopes.





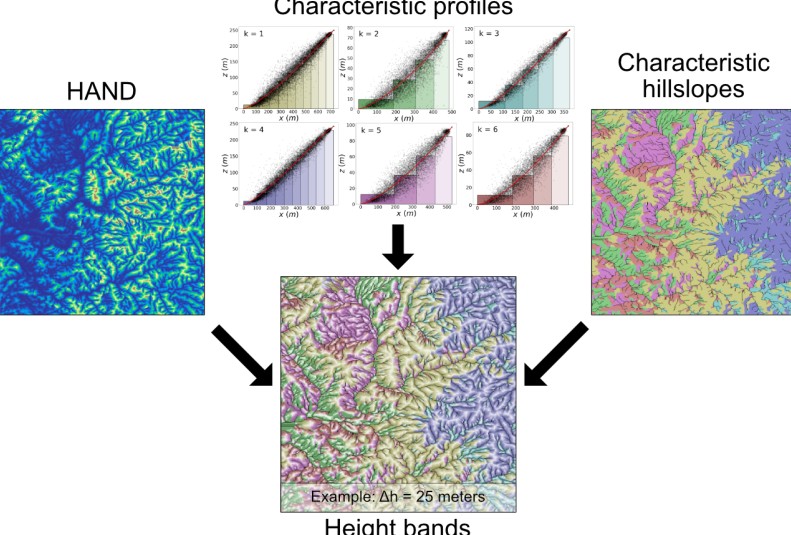

**Figure 4.** The profile of each characteristic hillslope is constructed by fitting $h = H\left[1 - \left(1 - (x/L)^a\right)^b\right]$ to the combined profiles of all its corresponding hillslopes. The characteristic profile is then discretized into $\lceil H_i/\Delta h \rceil$ height bands. Finally, the HAND and characteristic hillslope maps are combined with the discretized profiles to assign a unique height band to each fine-scale grid cell. In the height bands image, each band of each characteristic hillslope is represented by a different color.





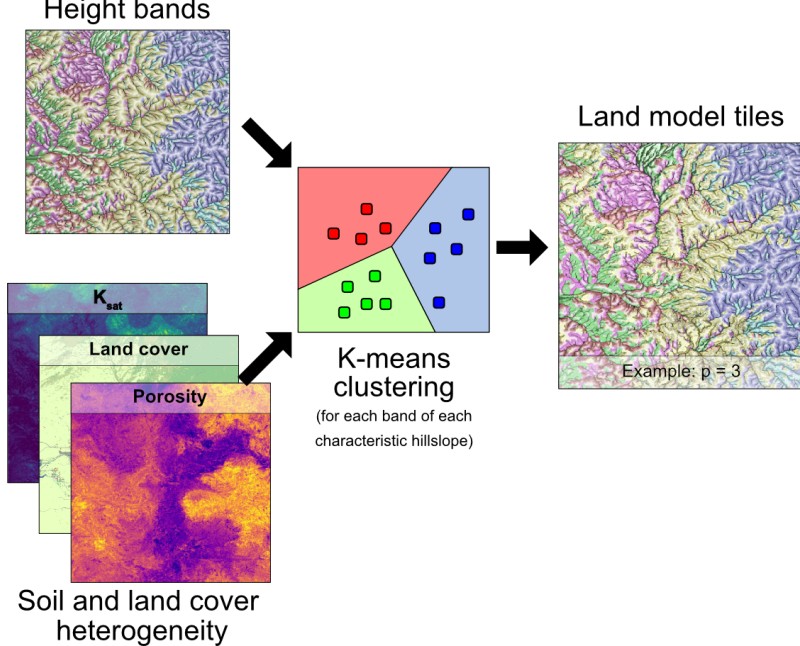

**Figure 5.** For each height band of each characteristic hillslope, the corresponding fine-scale grid cells are clustered into $p$ intra-band tiles according to their land cover and soil properties.





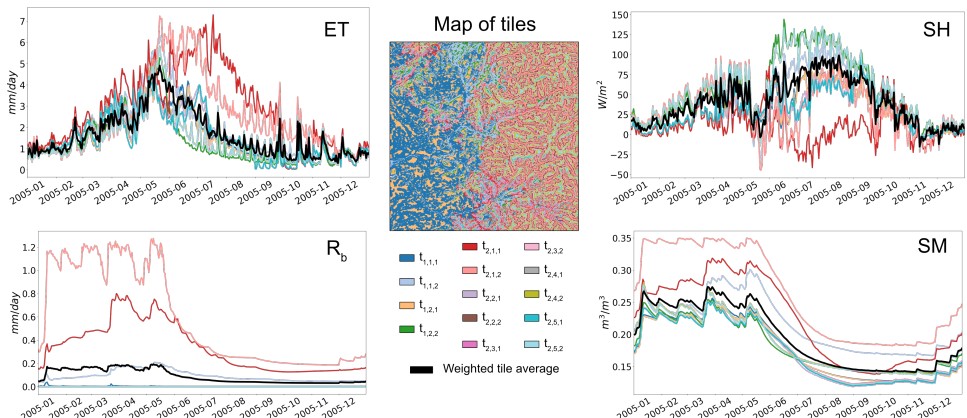

**Figure 6.** As an exploratory simulation, the LM4-HB model is run between 2002 and 2014 using 14 tiles assembled via HMC. The tile simulations are shown for evapotranspiration (ET), sensible heat flux (SH), baseflow ($R_b$), and root zone soil moisture (SM) for 2005; the time series are color-coded to correspond with the 30-meter map of tiles. Each tile has a corresponding id $t_{i,j,k}$, where $i$ is the characteristic hillslope, $j$ is the height band, and $k$ is the intra-band cluster. For each variable, the tile weighted average time series (macroscale estimate) is superimposed on the tile simulations for comparison.





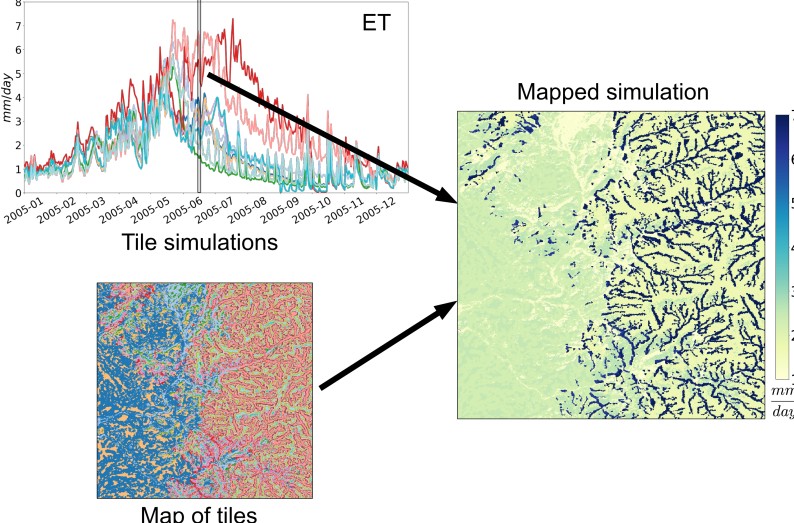

**Figure 7.** Using the exploratory simulation, the tile simulated daily evapotranspiration (ET) values for June 16th, 2005 are mapped out onto the 30-meter fully distributed grid using the HMC-assembled fine-scale map of tiles.





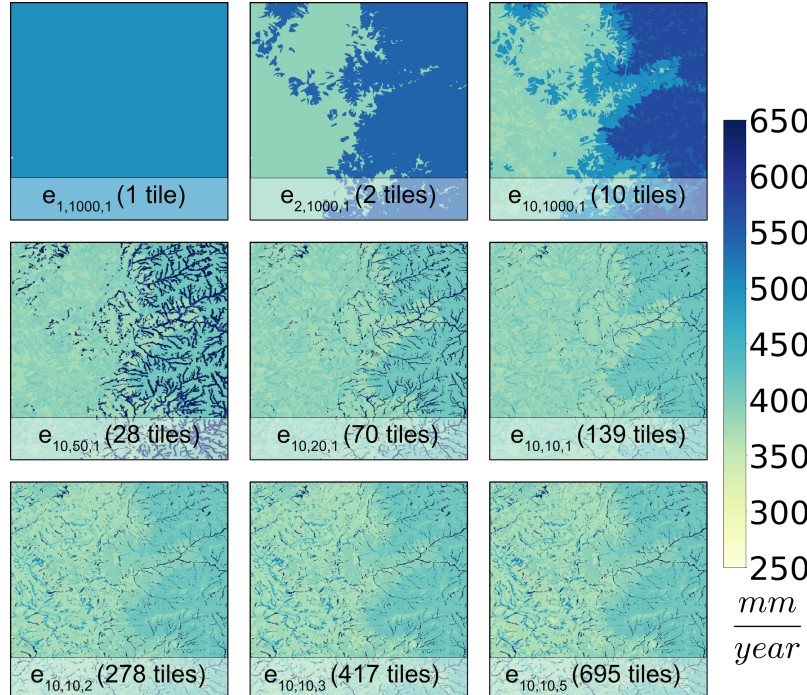

**Figure 8.** Visual comparison of the mapped out annual mean (2002-2014) of simulated evapotranspiration for the 9 model experiments in Table 1.




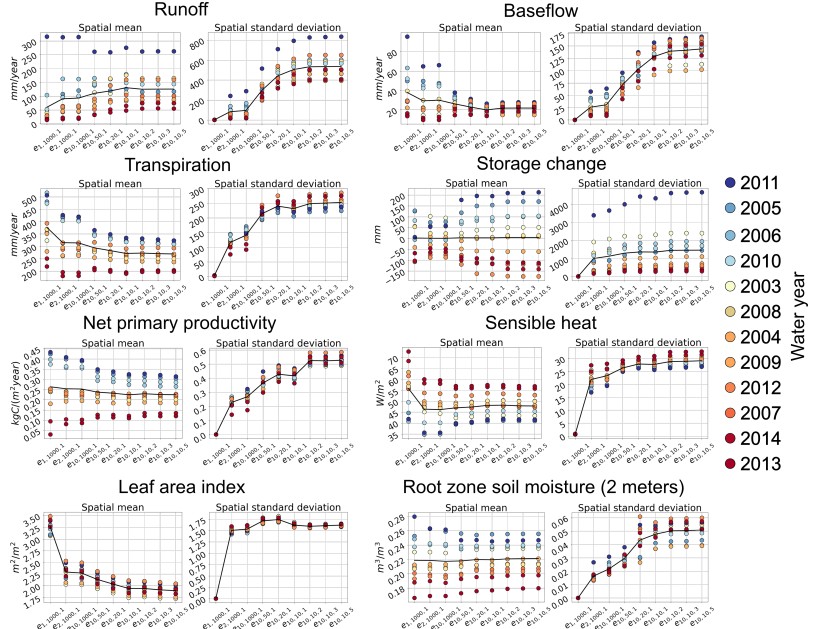

**Figure 9.** Comparison of the model experiments in Table 1. For each experiment, the spatial mean and spatial standard deviation for all water years (October 1st - September 30th) between 2003 and 2014 are plotted for a suite of states and fluxes. The corresponding values for each water year are color-coded according to their precipitation. The black line shows the annual mean.



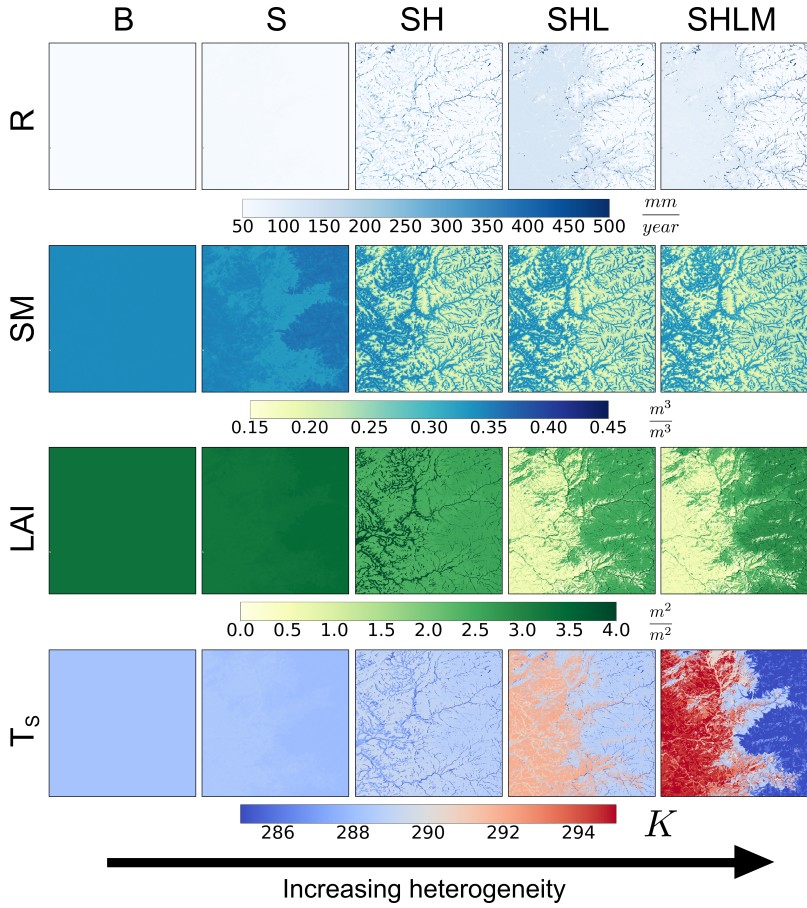

**Figure 10.** Visual comparison of the mapped out annual mean (2002-2014) of simulated runoff (R), 2 meter root zone soil moisture (SM), leaf area index (LAI), and soil temperature ($T_s$) for the 5 model experiments in Table 2.



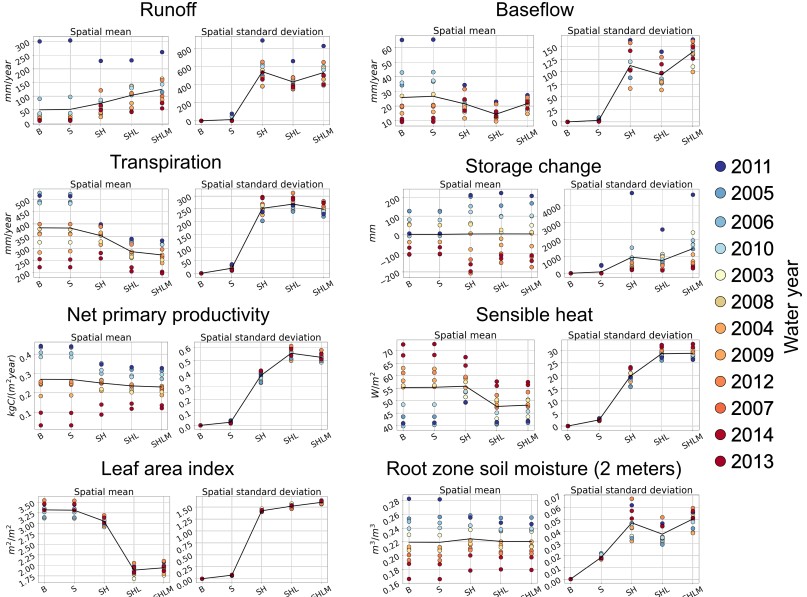

**Figure 11.** Comparison of the model experiments in Table 2. For each experiment, the spatial mean and spatial standard deviation for all water years between 2003 and 2014 are plotted for a suite of states and fluxes. The corresponding values for each water year are color-coded according to their precipitation. The black line shows the annual mean.



**Table 1.** Through a series of model experiments, the HMC parameters are adjusted to assess their role in the modeled heterogeneity. $e_{k,\Delta h,p}$ is the experiment id, $k$ is the number of characteristic hillslopes, $\Delta h$ is the difference in height between adjacent height bands, $p$ is the number of intra-band clusters, and $n_{tiles}$ is the resulting number of tiles.

| $e_{k,\Delta h,p}$ | $k$ | $\Delta h$ | $p$ | $n_{tiles}$ |
|---|---|---|---|---|
| $e_{1,1000,1}$ | 1 | 1000 | 1 | 1 |
| $e_{2,1000,1}$ | 2 | 1000 | 1 | 2 |
| $e_{10,1000,1}$ | 10 | 1000 | 1 | 10 |
| $e_{10,50,1}$ | 10 | 50 | 1 | 28 |
| $e_{10,20,1}$ | 10 | 20 | 1 | 70 |
| $e_{10,10,1}$ | 10 | 10 | 1 | 139 |
| $e_{10,10,2}$ | 10 | 10 | 2 | 278 |
| $e_{10,10,3}$ | 10 | 10 | 3 | 417 |
| $e_{10,10,5}$ | 10 | 10 | 5 | 695 |





**Table 2.** Through a series of model experiments, the heterogeneity of model parameters and forcing data are turned on (heterogeneous) and off (homogeneous). For simplicity, the parameters and forcing data are grouped into soil, hillslope, land cover, and meteorology groups.

| $e_{id}$ | Soil | Hillslope | Land cover | Meteorology |
|---|---|---|---|---|
| B | Off | Off | Off | Off |
| S | On | Off | Off | Off |
| SH | On | On | Off | Off |
| SHL | On | On | On | Off |
| SHLM | On | On | On | On |





*Acknowledgements.* We appreciate M. Lee and K. Findell for reviewing and providing comments on the manuscript. The statements, findings, conclusions, and recommendations are those of the authors and do not necessarily reflect the views of the National Oceanic and Atmospheric Administration or the U.S. Department of Commerce.



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
