# Peer review of "Harnessing Big Data to Rethink Land Heterogeneity in Earth System Models"

_Hydrology and Earth System Sciences, 2017_

## Referee Comment (RC1) · K. J. Beven (Referee) · 9 Nov 2017

I do not want to be too critical because I fully understand why this work is being done and so much effort has gone into it, and I was arguing for the inclusion of more heterogeneity into land surface parameterisations more than 20 years ago. But it is also an excellent reminder to me as to why I have chosen not to work in this area - the underlying "science" is really not very satisfying. There is an implicit assumption throughout the paper that the fine scale model, based on the various databases available, is correct. Further results are essentially expressed relative to this fine scale model and show that it is indeed important to account for heterogeneity (to the extent of the order of 300 deterministic tiles per grid square). But how can we still ignore the difficulties of parameterising the relevant processes and the resulting uncertainties in effective parameter values and how that might impact on the appropriate complexity of models to be considered. There are much simpler ways of incorporating heterogeneity into such predictions – is the use of 300 deterministic tiles really the best strategy to match actual landscape scale fluxes and integral measurements such as stream discharge? And to ensure that such heterogeneities are reflected in longer term future predictions?

So I would suggest that this is a paper that is acceptable with some minor modifications, but it is quite the wrong paper for what is needed.

Some specific comments.

P2 L5 Beven and Kirkby might be a relevant reference but not in this context - it did not deal in any way with large scale hydrology.

But the next sentence is also wrong – TOPMODEL was designed with a view to handling sub-grid variability in hydrology that would then form the basis for other predictions (including land management effects) – we were both interested in predicting sediment transport at the time. I also later extended it to allow for variable infiltration and conductivity characteristics and more explicit surface energy balance calculations. Also same issue on P12 L7

P2 L30 Beven and Kirkby did not use a mosaic approach – though there were later forms of TOPMODEL that did so, including TOPLATS (developed at Princeton!!!) and later Dynamic TOPMODEL based on multiple overlays and which forms the basis for HydroBlocks.

P9 L1 Why is this expected? Upslope elements surely contribute "baseflow" to those tiles adjacent to channels? Or do you need to explain what you mean by baseflow (there is an interaction here with your definition of channels – there would be many downslope fluxes in small, possibly ephemeral streams at scales much less than 100000km^2)?

P9 L18 patterns in ET. This is also evident from remote sensing derived estimates –

but there are two aspects to this – one Is the uncertainty associated with both derived and modelled estimates (see papers by Franks et al. e.g. WRR 1997, 1999), the other is whether the type of model being used here can really be supported by the data (see Bashford et al. HP2002). Both issues are worthy of mention given that this paper considers only a comparison of model runs from a single structure without uncertainty.

P12 L23. This is a claim too far. There has been a lot of previous work linking topography, hydrology and energy balance/evapotranspiration and vegetation interactions even if not such detail (e.g. Franks et al.; Quinn et al HESS, 1998; Blazkova et al. WRR 2002). Then there are all the Teague and Band RHESSYS papers, and the Topmodel based land surface scheme used by MeteoFrance (ISBA-TOPMODEL, Vincendon et al JH2010). . .. And more!!

P12 L28 It is clear that these heterogeneities make a difference, and that there are more that we can think of, including the biophysical feedbacks, that would undoubtedly result in more complex parameterisations. BUT, in complex terrain with heterogeneous cover that also introduces all sort of local boundary layer effects that have proven too difficult to deal with at the micrometeorological scale. And yet, at the landscape scale the actual heterogeneous latent heat and gas fluxes are additive, effectively filtering out much of the variability (the REA idea that the distribution might be important but the pattern may not be) – so is there not an issue of appropriate complexity of model constructs that needs to be addressed here (e.g. Bashford et al HP2002) – especially given the uncertainty with which the parameters of your model structure can be defined.

P14 L17. But that only considers fitting one model to another. How can you simply assume that your fine scale model is correct. It is not (no small scale channels, preferential flows, lack of knowledge of role of regolith and geology, parameterisations of biogeochemical processes etc) or at best involves significant uncertainty in effective parameter values – it does not consider whether the underlying model structure and parameters are adequate to reproduce observations, and what those observations might mean in this context (discharges, FLUXNET, surface temperature images,. . ...).

Section 5.3 is really addressing the wrong problem.

---

## Referee Comment (RC2) · Anonymous Referee #2 · 6 Dec 2017

Authors have implemented a hierarchical multivariate clustering approach to represent land surface heterogeneity in an Earth System Model Grid cell. The approach is taking advantage of fine resolution datasets from various sources in addition to a Digital Elevation Model (DEM) to identify a set of characteristics hillslopes through a multivariate clustering algorithm. Each characteristic hillslope is then discretized into height bands, and a set of tiles are delineated for each band to represent heterogeneity of soil and land cover types.

While the topic of this paper is of great interest for the global hydrologic modeling community, no attempt has been made to assess the performance of these simulations against observations. Indeed, fine resolution model simulations are used as the truth. I suggest authors to take advantage of various satellite products to define what scale of

heterogeneity needs to be incorporated in each ESM grid cell to better represent land surface states and fluxes.

Authors have performed a limited sets of sensitivity analysis to assess the impacts of height band length and number of clusters on simulated fluxes. However, no insights have been provided in order to define these parameters in various landscapes. Further discussion will be really helpful to inform the modelers.

No physically based approach are presented for delineating the height bands. Can authors implement a more physically based approach for defining the height bands? For example, the methodology of Khan et al. 2013 (Environmental Modelling and Software) for delineating landforms can be informative in this case.

Another issue is regarding the number of variables that are required for a multivariate cluster analysis. I suggest authors to perform a sensitivity analysis to identify factors that are most influential in defining these clusters.

Further descriptions about the discretization methodology is required particularly those that relate to Figure 4. I suggest authors to prepare a flowchart that explains every step from discretization to mapping back the results to the fine resolution.

Further clarification regarding the simulation elements are required.

Another major point is how does the surface and subsurface connectivity between the tiles within a single band and between bands are maintained? What rules do you implement?

How tiles are represented for model simulations? Does each tile represented by a point scale simulation?

P6-L20: The authors indicate that attributes of each characteristic hillslope is obtained through arithmetic averaging. How do you handle categorical data like soil type and vegetation types in this case?

[Figure]

Section 3.2.1. What is a typical size of a characteristic hillslope?

Section 3.2.2. L 30: Needs further explanation.

Can authors provide further insights and general recommendations for implementation of their approach in other geographic regions?

Can authors provide further information about the enhancements made in this approach compared to the earlier work of delineating HydroBlocks?

―――――――――――――――――――

---

## Author Comment (AC1) · 27 Dec 2017

**Response to reviewers' comments**

"Harnessing Big Data to Rethink Land Heterogeneity in Earth System Models" by N. W. Chaney, M. H. J. Van Huijgevoort, E. Shevliakova, S. Malyshev, P. C. D. Milly, P. P. G. Gauthier, B. N. Sulman

We thank the reviewers for their time and helpful comments. We have addressed each point below. Reviewer comments are shown in *blue italics*, while author responses are shown in unformatted text.

**Reviewer #1**: *I do not want to be too critical because I fully understand why this work is being done and so much effort has gone into it, and I was arguing for the inclusion of more heterogeneity into land surface parameterisations more than 20 years ago. But it is also an excellent reminder to me as to why I have chosen not to work in this area - the underlying "science" is really not very satisfying.*

*There is an implicit assumption throughout the paper that the fine scale model, based on the various databases available, is correct. Further results are essentially expressed relative to this fine scale model and show that it is indeed important to account for heterogeneity (to the extent of the order of 300 deterministic tiles per grid square). But how can we still ignore the difficulties of parameterising the relevant processes and the resulting uncertainties in effective parameter values and how that might impact on the appropriate complexity of models to be considered.*

We thank the reviewer for this feedback. The main purpose of this paper is to illustrate how the information content of the multi-scale heterogeneity information from satellite remote sensing can be effectively and efficiently incorporated into large-scale models; this is accomplished by taking advantage of the correlation between the drivers of spatial heterogeneity via similarity concepts. This paper aims to show how hierarchical multivariate clustering enables large-scale models to characterize multi-scale land heterogeneity without having to resort to a more brute force approach (i.e., fully distributed model). We acknowledge that the deterministic 30-meter fully distributed model should not be the end goal—the uncertainty in the model structure, parameters, and forcing at those scales will quickly minimize any added benefit of modeling at higher spatial scales. The end goal is to provide an approach that allows one to obtain the same benefits of the fully distributed model (e.g., field-scale validation) while ensuring sufficient efficiency to allow for large ensembles to account for the different sources of uncertainty. We will clarify this point in the revised manuscript.

*There are much simpler ways of incorporating heterogeneity into such predictions – is the use of 300 deterministic tiles really the best strategy to match actual landscape scale fluxes and integral measurements such as stream discharge? And to ensure that such heterogeneities are reflected in longer-term future predictions?*

One of the principal challenges of coupling the water, energy, and carbon cycles in land models is that the multi-scale heterogeneity of the physical environment can, at times, have different importance for each cycle. Therefore, by focusing exclusively on the hydrologic cycle, there will be cases where the heterogeneity of the other processes will be disregarded. Until now, land models have addressed this issue by tying together different parameterizations of heterogeneity while disregarding the inherent covariance of these different cycles (e.g., applying the same variable infiltration capacity curve for each land cover type). The goal in this study is to develop an approach that places all these sources of heterogeneity into a single n-dimensional space from which the characteristic tiles are computed. We acknowledge that the result of 300 tiles will be site dependent and will vary depending on model structure, parameters, and forcing. A preliminary implementation of this approach globally illustrates that in most cases you need much fewer than 300 tiles. We will add this to the discussion in the revised manuscript.

*P2 L5 Beven and Kirkby might be a relevant reference but not in this context - it did not deal in any way with large scale hydrology. But the next sentence is also wrong – TOPMODEL was designed with a view to handling sub-grid variability in hydrology that would then form the basis for other predictions (including land management effects) – we were both interested in predicting sediment transport at the time. I also later extended it to allow for variable infiltration and conductivity characteristics and more explicit surface energy balance calculations. Also same issue on P12 L7 P2 L30 Beven and Kirkby did not use a mosaic approach – though there were later forms of TOPMODEL that did so, including TOPLATS (developed at Princeton!!!) and later Dynamic TOPMODEL based on multiple overlays and which forms the basis for HydroBlocks.*

We thank the reviewer for pointing out the important issues with the references in the manuscript. We will revise the references accordingly in the updated manuscript.

*P9 L1 Why is this expected? Upslope elemeents surely contribute "baseflow" to those tiles adjacent to channels? Or do you need to explain what you mean by base- flow (there is an interaction here with your definition of channels – there would be many downslope fluxes in small, possibly ephemeral streams at scales much less than 100000kmˆ2)?*

In the original manuscript, by baseflow we meant to refer to the flow of subsurface water from the bottom of the hillslope to the channel. We now understand how the existing text is confusing given that there are cases where subsurface runoff can emerge in intermediate tiles even though that does not occur in the shown time series. The updated manuscript will clarify this sentence.

*P9 L18 patterns in ET. This is also evident from remote sensing derived estimates –but there are two aspects to this – one Is the uncertainty associated with both derived and modelled estimates (see papers by Franks et al. e.g. WRR 1997, 1999), the other is whether the type of model being used here can really be supported by the data (see Bashford et al. HP2002). Both issues are worthy of mention given that this paper considers only a comparison of model runs from a single structure without uncertainty.*

Thank you for this suggestion. One of the primary outcomes of this work is the ability to more critically assess what land models actually suggest is happening at the landscape scale. By mapping out the results, it makes it possible to make better use of the range of measurements (field-scale to macro-scale). However, the maps are still model products and thus although visually appealing they could be further from reality than a homogeneous map. Although addressed briefly in section 5.2, we will update and expand this issue in the updated manuscript.

*P12 L23. This is a claim too far. There has been a lot of previous work linking topography, hydrology and energy balance/evapotranspiration and vegetation interactions even if not such detail (e.g. Franks et al.; Quinn et al HESS, 1998; Blazkova et al. WRR 2002). Then there are all the Teague and Band RHESSYS papers, and the Topmodel based land surface scheme used by MeteoFrance (ISBA-TOPMODEL, Vincendon et al JH2010). . .. And more!!*

We appreciate the reviewer for catching this overreaching statement in the original manuscript. The intended purpose of that statement was to mention that this study was the first time to the authors' knowledge that this level of land heterogeneity has been included in Earth system models for use in seasonal to climate scale predictions. We will update this statement in the revised manuscript.

*P12 L28 It is clear that these heterogeneities make a difference, and that there are more that we can think of, including the biophysical feedbacks, that would undoubtedly result in more complex parameterisations. BUT, in complex terrain with heterogeneous cover that also introduces all sort of local boundary layer effects that have proven too difficult to deal with at the micrometeorological scale. And yet, at the landscape scale the actual heterogeneous latent heat and gas fluxes are additive, effectively filtering out much of the variability (the REA idea that the distribution might be important but the pattern may not be) – so is there not an issue of appropriate complexity of model constructs that needs to be addressed here (e.g. Bashford et al HP2002) – especially given the uncertainty with which the parameters of your model structure can be defined.*

One of the persistent challenges within land-atmosphere coupled models (especially over complex terrain) is to determine how to represent statistically the inherent heterogeneity that emerges as a result of the two-way interaction of land and the boundary layer. This becomes even more challenging when considering that the boundary layer of the tiles should also interact dynamically. We agree that these

issues tied to the underlying uncertainty of the input data can lead to uncertain benefits due to the emerging additional strong uncertainties. However, the authors view the method presented as a robust path forward that capitalizes on the existing petabytes of environmental information. In any case, we acknowledge the inherent benefits of using REAs and can visualize using them to synthesize and expand the similarity concepts explored in this study. These issues will be mentioned and addressed within the discussion section of the revised manuscript.

*P14 L17. But that only considers fitting one model to another. How can you simply assume that your fine scale model is correct. It is not (no small scale channels, preferential flows, lack of knowledge of role of regolith and geology, parameterisations of biogeochemical processes etc) or at best involves significant uncertainty in effective parameter values – it does not consider whether the underlying model structure and parameters are adequate to reproduce observations, and what those observations might mean in this context (discharges, FLUXNET, surface temperature images,. . ...). Section 5.3 is really addressing the wrong problem.*

As addressed above, the purpose of this study is not to suggest that the fully distributed version of LM4-HB is "true". Instead, it is meant to illustrate a path towards ensuring that the multi-scale heterogeneity information that emerges from using the very high-resolution information in the brute force method (fully distributed model) is represented robustly in the semi-distributed model. In reality, the methods introduced in this study (HMC) are meant to be flexible to different model structures and parameter values. We envision an ensemble of tile structures being constructed per macroscale grid cell. This would ensure that any added benefit of fully distributed models would be appropriately accounted for in the semi-distributed model while allowing for the underlying benefits of the reduced order model to make it possible to run large ensembles to constrain the unavoidable uncertainty. We will update section 5.3 to expand on this issue and to emphasize the intended nature of this work to be used for large ensemble frameworks.

**Reviewer #1**: *Authors have implemented a hierarchical multivariate clustering approach to represent land surface heterogeneity in an Earth System Model Grid cell. The approach is taking advantage of fine resolution datasets from various sources in addition to a Digital Elevation Model (DEM) to identify a set of characteristics hillslopes through a multivariate clustering algorithm. Each characteristic hillslope is then discretized into height bands, and a set of tiles are delineated for each band to represent heterogeneity of soil and land cover types.*

*While the topic of this paper is of great interest for the global hydrologic modeling community, no attempt has been made to assess the performance of these simulations against observations.*

We thank the reviewer for this feedback. The main purpose of this work is to develop an approach to illustrate how high-resolution satellite information can be harnessed to provide an efficient and effective representation of multi-scale heterogeneity in macroscale land surface and hydrologic models. As such it is meant to be primarily a technical paper that illustrates how this can be accomplished. A follow-up paper that is currently being written takes HMC and applies it over the globe for each 1-degree land grid cell. The global results are currently being thoroughly evaluated using a suite of observations including streamflow, MODIS LAI, and FLUXNET. For the sake of brevity and conciseness we have left the comprehensive evaluation for that subsequent study. We will mention this subsequent paper and its role in evaluation in the updated manuscript's discussion section.

*Indeed, fine resolution model simulations are used as the truth. I suggest authors to take advantage of various satellite products to define what scale of heterogeneity needs to be incorporated in each ESM grid cell to better represent land surface states and fluxes.*

The fine-scale model can be seen as the brute-force method that ensures that the multi-scale heterogeneity information in the satellite remote sensing is accounted for in the model. The goal in HMC is to closely approximate the level of detail that would be possible through a fully distributed model with a fraction of the computation. Given that the fully distributed model will also be wrong due to structural and parameter uncertainties, the approach here is meant to be able to find the best of both worlds: allow for large ensemble frameworks to constrain these unavoidable uncertainties while ensuring the robust characterization of the observed multi-scale heterogeneity.

*Authors have performed a limited sets of sensitivity analysis to assess the impacts of height band length and number of clusters on simulated fluxes. However, no insights have been provided in order to define these parameters in various landscapes. Further discussion will be really helpful to inform the modelers.*

HMC has recently been applied over the globe at a 1-degree grid cell. The required difference in elevation between adjacent height bands ensures a different number of tiles are required per grid cell; grid cells in flat regions need between 5-20 while those in mountainous regions require between 100-400. The subsequent evaluation paper will present these results. However, even this subsequent study uses a predefined set of parameters for HMC. For this purpose, as outlined in section 5.3, future work should explore multiple approaches to determine how to compute the optimal parameters per grid cell. Possible approaches include regionalizing optimized HMC parameters using existing macroscale environmental information.

*No physically based approach are presented for delineating the height bands. Can authors implement a more physically based approach for defining the height bands?*

*For example, the methodology of Khan et al. 2013 (Environmental Modelling and Software) for delineating landforms can be informative in this case.*

We thank the reviewer for this important feedback. To ensure the height bands are connected to the observed landscape, the height above nearest drainage area is calculated from the high-resolution elevation data and the computed channel network. The combination of the map of hillslopes and the HAND map are then used to compute the characteristic of each hillslopes (e.g., slope). After clustering the hillslopes using these computed properties, each characteristic hillslope is discretized into $n$ height bands where $n$ is the rounded u up value of the HAND value at the ridge for a given characteristic hillslope divided by a user-defined parameter $\Delta h$ that is the difference between adjacent height bands. These height bands are then connected back to the original map to provide a high-resolution representation as shown in Figure 4. That being said, we agree that computing the channel network, delineating the hillslopes, and discretizing the hillslopes will play a critical role in the end result and can be itself uncertain; as a result, this will be an important area of further research. In the revised manuscript, we will include a new discussion section that provides an overview of the importance of appropriately delineating and discretizing the hillslopes and the need for further research within this area.

*Another issue is regarding the number of variables that are required for a multivariate cluster analysis. I suggest authors to perform a sensitivity analysis to identify factors that are most influential in defining these clusters.*

We agree that the clustering analysis will be susceptible to the number of variables used; if the correlation between many of the variables is high, the additional information provided by the increase in variables will be relatively small. Although a concerted effort was made to ensure the chosen environmental properties were sufficiently independent, further work should be done along this line. One example that is being explored by one of the authors is to use principal component analysis to arrive at the most significant features. Another approach is to use an iterative approach to assign different weights to the different input data. In any case, this is beyond the scope of this study and should be addressed in future research. That being said, recognizing its importance, it will be discussed within the new discussion section in the revised manuscript on improving HMC.

*Further descriptions about the discretization methodology is required particularly those that relate to Figure 4. I suggest authors to prepare a flowchart that explains every step from discretization to mapping back the results to the fine resolution. Further clarification regarding the simulation elements are required.*

Figures 3 through 5 illustrate the different steps taken in the hierarchical multivariate clustering algorithm. We believe that these figures are sufficiently detailed and do not require further additions. However, we understand the reviewer's concern with understanding the discretization processes and will expand section 3.2.2 and the caption of Figure 4 in the revised manuscript for clarification.

*Another major point is how does the surface and subsurface connectivity between the tiles within a single band and between bands are maintained? What rules do you implement?*

The focus of this study is on the development and implementation of the hierarchical multivariate clustering algorithm. For simplicity, we do not provide a detailed explanation of the hillslope model which is explained in more detail in [*Subin et al.*, 2014]. However, the main idea is that all tiles interact via the subsurface via the exchange of water through Richards' equation. The parameter used for the interaction between height bands is the interface width between two adjacent height bands, the length between the center points of the height bands, and the effective hydraulic conductivity between the two. The exchange of subsurface water occurs horizontally between the different layers of each adjacent height band. A similar approach is used for the intra-height tiles except in this case the interface width is fixed by a user-defined parameter. Currently, surface runoff is routed instantaneously to the channel at the base of the hillslope. Future work could explore implementing a surface routing scheme as well; although it is unclear if the added complexity and computation time would lead to sufficiently beneficial differences.

*How tiles are represented for model simulations? Does each tile represented by a point scale simulation?*

As the reviewer suggests, each tile simulation can be interpreted as a point simulation that is scaled up to the areal coverage of the tile. However, it would be more appropriate to see each tile as a field-scale simulation since many of the parameterized processes (e.g., vegetation dynamics) are more suitable for interpretation above ~10 meter scales. These tiles then interact via the subsurface along the discretized hillslope.

*P6-L20: The authors indicate that attributes of each characteristic hillslope is obtained through arithmetic averaging. How do you handle categorical data like soil type and vegetation types in this case?*

We thank the reviewer to noticing this issue. The clustering analysis is performed on only continuous data and thus is not susceptible to issues with categorical data. However, when providing model parameters for LM4-HB some properties such as plant species needs to be defined as a categorical type. In this case, after constructing the map of tiles at 30 meters, the mode of each categorical type for each tile calculated from the 30-meter map of plant species is used as the tile's species. This will mean that some information is lost; however, from experience, statistically, the result ends up being satisfactory as long there are a sufficient number of tiles. We will clarify this in the updated manuscript.

*Section 3.2.1. What is a typical size of a characteristic hillslope?*

From our analysis for this domain and other regions, the typical length of each hillslope ranges between 200 to 2000 meters with slopes ranging between 0.01 and 0.5.

*Section 3.2.2. L 30: Needs further explanation.*

We appreciate the reviewer's feedback on the need for clarification within the hillslope discretization section. We will ensure that the revised manuscript addresses this issue in further detail.

*Can authors provide further insights and general recommendations for implementation of their approach in other geographic regions?*

Although not mentioned explicitly in the original manuscript, ongoing work has used globally available environmental data to implement HMC over the globe. Section 5.3 in the manuscript discusses how future work could explore how to arrive at the optimal HMC parameters per macroscale grid cell over the globe.

*Can authors provide further information about the enhancements made in this approach compared to the earlier work of delineating HydroBlocks?*

The approach used originally in the HydroBlocks paper [*Chaney et al.*, 2016] can be seen as a "brute force" method that uses multivariate clustering without an explicit accounting of the hydrologic structure. This leads to connections between tiles or HRUs that at times can be unrealistic and lead to poor performance (e.g., subsurface runoff). This paper aims to take the ideas explored in HydroBlocks a step further and make use of common hydrologic knowledge to create a hierarchy of tiles that takes advantage of the clustering approach while ensuring a hydrologically consistent system. As shown in Figure 9, this leads to the convergence of baseflow as the number of tiles increases which is not the case in the HydroBlocks paper.

We would again like to thank both reviewers for their time and helpful comments.

**References**

Chaney, N., P. Metcalfe, and E. F. Wood (2016), HydroBlocks: A Field-scale Resolving Land Surface Model for Application Over Continental Extents, *Hydrol. Process.*, doi:10.1002/hyp.10891.

Subin, Z. M., P. C. D. Milly, B. N. Sulman, S. Malyshev, and E. Shevliakova (2014), Resolving terrestrial ecosystem processes along a subgrid topographic gradient for an earth-system model, *Hydrol. Earth Syst. Sci. Discuss.*, *11*(7).

---

## Author Response (AR1)

**Response to reviewers' comments**

"Harnessing Big Data to Rethink Land Heterogeneity in Earth System Models" by N. W. Chaney, M. H. J. Van Huijgevoort, E. Shevliakova, S. Malyshev, P. C. D. Milly, P. P. G. Gauthier, B. N. Sulman

We thank the reviewers for their time and helpful comments. We have addressed each point below. Reviewer comments are shown in *blue italics*, while author responses are shown in unformatted text.

**Editor**: *Thank you for your interests in HESS. We received review comments from two qualified reviewers. Both acknowledged the importance of the topic, but pointed out places that require clarifications and justifications. For example, reviewer #2 mentioned "I suggest authors take advantage of various satellite products to define what scale of heterogeneity needs to be incorporated in each ESM grid cell to better represent land surface states and fluxes." This would need a more direct response. In addition, you mentioned that "As such it is meant to be primarily a technical paper" in your response. Please make sure to select the appropriate manuscript type during revision submission (e.g., technical notes)*

We thank the editor for these recommendations. As suggested, we have addressed the comment from reviewer #2 regarding the use of satellite data to define the heterogeneity explicitly in both the response and the revised manuscript.

Although the paper has an important focus on methods, we argue that it is still primarily a research paper because it addresses a key challenge in hydrology to more fully characterize multi-scale heterogeneity in global hydrologic and land surface models. Furthermore, it provides a key finding that including heterogeneity leads to a dampening of the hydrologic response. For these reasons we have resubmitted the manuscript as a research paper.

**Reviewer #1**: *I do not want to be too critical because I fully understand why this work is being done and so much effort has gone into it, and I was arguing for the inclusion of more heterogeneity into land surface parameterisations more than 20 years ago. But it is also an excellent reminder to me as to why I have chosen not to work in this area - the underlying "science" is really not very satisfying.*

*There is an implicit assumption throughout the paper that the fine scale model, based on the various databases available, is correct. Further results are essentially expressed relative to this fine scale model and show that it is indeed important to account for heterogeneity (to the extent of the order of 300 deterministic tiles per grid square). But how can we still ignore the difficulties of parameterising the relevant processes and the resulting uncertainties in effective parameter values and how that might impact on the appropriate complexity of models to be considered.*

We thank the reviewer for this feedback. The main purpose of this paper is to illustrate how the information content of the multi-scale heterogeneity information

from satellite remote sensing can be effectively and efficiently incorporated into large-scale models; this is accomplished by taking advantage of the correlation between the drivers of spatial heterogeneity via hydrologic similarity. This paper aims to show how hierarchical multivariate clustering enables large-scale models to characterize multi-scale land heterogeneity without having to resort to more brute force approaches (i.e., fully distributed model). We acknowledge that the deterministic 30-meter fully distributed model should not be the end goal—the large uncertainty in model structure, parameters, and forcing will quickly minimize any added benefit of modeling at higher spatial scales. Instead, the end goal is to provide an approach that robustly accounts for the different sources of multi-scale heterogeneity while ensuring simplicity and computational efficiency. In practice, this makes it possible to minimize the scaling uncertainties and instead focus on the actual process representation. We have added this clarification in the discussion section of the revised manuscript.

*There are much simpler ways of incorporating heterogeneity into such predictions – is the use of 300 deterministic tiles really the best strategy to match actual landscape scale fluxes and integral measurements such as stream discharge? And to ensure that such heterogeneities are reflected in longer-term future predictions?*

One of the principal challenges of coupling the water, energy, and carbon cycles in land models is that the multi-scale heterogeneity of the physical environment can, at times, have different importance for each cycle. Therefore, by focusing exclusively on the hydrologic cycle, there will be cases where the heterogeneity of the other processes will be disregarded. Until now, land models have addressed this issue by tying together different parameterizations of heterogeneity while disregarding the inherent covariance of these different cycles (e.g., applying the same variable infiltration capacity curve for each land cover type). The goal in this study is to develop an approach that places all these sources of heterogeneity into a single n-dimensional space from which the characteristic tiles are computed. We acknowledge that the result of 300 tiles will be site and application dependent and will vary depending on model structure, parameters, and forcing. We have added this to the discussion in the revised manuscript.

*P2 L5 Beven and Kirkby might be a relevant reference but not in this context - it did not deal in any way with large scale hydrology. But the next sentence is also wrong – TOPMODEL was designed with a view to handling sub-grid variability in hydrology that would then form the basis for other predictions (including land management effects) – we were both interested in predicting sediment transport at the time. I also later extended it to allow for variable infiltration and conductivity characteristics and more explicit surface energy balance calculations. Also same issue on P12 L7 P2 L30 Beven and Kirkby did not use a mosaic approach – though there were later forms of TOPMODEL that did so, including TOPLATS (developed at Princeton!!!) and later Dynamic TOPMODEL based on multiple overlays and which forms the basis for*

*HydroBlocks.*

We thank the reviewer for pointing out our misinterpretation of the connection of the original TOPMODEL paper with this work and large-scale hydrology more generally. We have revised the references accordingly in the updated manuscript.

*P9 L1 Why is this expected? Upslope elemeents surely contribute "baseflow" to those tiles adjacent to channels? Or do you need to explain what you mean by base- flow (there is an interaction here with your definition of channels – there would be many downslope fluxes in small, possibly ephemeral streams at scales much less than 100000km^2)?*

By baseflow we mean the fraction of streamflow that comes from subsurface runoff. Although LM4-HB almost exclusively produces baseflow at the hillslope/channel interface, we agree that the missing finer scale channels will play a key role in baseflow production. In the revised manuscript we address how this issue could be addressed moving forward by improving the representation of the fine scale channel networks using higher resolution DEMs.

*P9 L18 patterns in ET. This is also evident from remote sensing derived estimates –but there are two aspects to this – one Is the uncertainty associated with both derived and modelled estimates (see papers by Franks et al. e.g. WRR 1997, 1999), the other is whether the type of model being used here can really be supported by the data (see Bashford et al. HP2002). Both issues are worthy of mention given that this paper considers only a comparison of model runs from a single structure without uncertainty.*

Thank you for this suggestion. One of the primary outcomes of this work is the ability to more critically assess what land models actually suggest is happening at the field scale. By mapping out the results, it makes it possible to make better use of the range of measurements (field-scale to macro-scale). However, we recognize that the field-scale representation are still model representations and thus although visually they might make sense (e.g., higher ET in riparian zones) that does not ensure that they provide skillful simulations of the observed surface fluxes. Although addressed briefly in section 5.2 in the original manuscript, we have expanded on this issue in the discussion section in the updated manuscript.

*P12 L23. This is a claim too far. There has been a lot of previous work linking topography, hydrology and energy balance/evapotranspiration and vegetation interactions even if not such detail (e.g. Franks et al.; Quinn et al HESS, 1998; Blazkova et al. WRR 2002). Then there are all the Teague and Band RHESSYS papers, and the Topmodel based land surface scheme used by MeteoFrance (ISBA-TOPMODEL, Vincendon et al JH2010). . .. And more!!*

We appreciate the reviewer for catching this overreaching statement in the original manuscript. The intended purpose of that statement was to mention that this study was the first time to the authors' knowledge that this level of land heterogeneity has been included in Earth system models for use in seasonal to climate scale predictions for coupled water, energy, and carbon simulations. To avoid confusion we have removed this statement in the revised manuscript.

*P12 L28 It is clear that these heterogeneities make a difference, and that there are more that we can think of, including the biophysical feedbacks, that would undoubtedly result in more complex parameterisations. BUT, in complex terrain with heterogeneous cover that also introduces all sort of local boundary layer effects that have proven too difficult to deal with at the micrometeorological scale. And yet, at the landscape scale the actual heterogeneous latent heat and gas fluxes are additive, effectively filtering out much of the variability (the REA idea that the distribution might be important but the pattern may not be) – so is there not an issue of appropriate complexity of model constructs that needs to be addressed here (e.g. Bashford et al HP2002) – especially given the uncertainty with which the parameters of your model structure can be defined.*

One of the persistent challenges within land-atmosphere coupled models (especially over complex terrain) is to determine how to represent statistically the inherent heterogeneity that emerges as a result of the two-way interaction of land and the boundary layer. This becomes even more challenging when considering that the boundary layer of the tiles should also interact dynamically. We agree that these issues tied to the underlying uncertainty of the input data can lead to additional model complexities that will limit possible benefits over strongly heterogeneous regions.

*P14 L17. But that only considers fitting one model to another. How can you simply assume that your fine scale model is correct. It is not (no small scale channels, preferential flows, lack of knowledge of role of regolith and geology, parameterisations of biogeochemical processes etc) or at best involves significant uncertainty in effective parameter values – it does not consider whether the underlying model structure and parameters are adequate to reproduce observations, and what those observations might mean in this context (discharges, FLUXNET, surface temperature images,. . ...).  Section 5.3 is really addressing the wrong problem.*

As addressed above, the purpose of this study is not to suggest that the fully distributed version of LM4-HB is accurate. Instead, it is meant to illustrate a path towards ensuring that the multi-scale heterogeneity information that emerges from using the very high-resolution information in the brute force method (fully distributed model) is represented robustly in the semi-distributed model. In reality, the methods introduced in this study (HMC) are meant to be flexible to different model structures and parameter values. We envision an ensemble of tile structures

being constructed per macroscale grid cell. This would ensure that any added benefit of fully distributed models would be appropriately accounted for in the semi-distributed model while allowing for the underlying benefits of the reduced order model to make it possible to run large ensembles to constrain the unavoidable uncertainty. The new section 5.3 in the updated manuscript expands on this issue and explains the limitations of finding a single optimal tile configuration.

**Reviewer #2**: *Authors have implemented a hierarchical multivariate clustering approach to represent land surface heterogeneity in an Earth System Model Grid cell. The approach is taking advantage of fine resolution datasets from various sources in addition to a Digital Elevation Model (DEM) to identify a set of characteristics hillslopes through a multivariate clustering algorithm. Each characteristic hillslope is then discretized into height bands, and a set of tiles are delineated for each band to represent heterogeneity of soil and land cover types.*

*While the topic of this paper is of great interest for the global hydrologic modeling community, no attempt has been made to assess the performance of these simulations against observations.*

We thank the reviewer for this feedback. The main purpose of this work is to develop an approach to illustrate how high-resolution satellite information can be harnessed to provide an efficient and effective representation of multi-scale heterogeneity in macroscale land surface and hydrologic models. This is accomplished by assessing how a reduced-order model can approximate the fully distributed model. We agree that observations should be included in evaluating how an improved representation of heterogeneity impacts the model skill. A follow-up paper that is currently being written takes HMC and applies it over the globe for each 1-degree land grid cell. The global results are currently being thoroughly evaluated using a suite of observations including GRDC streamflow, MODIS LAI, and FLUXNET. We have mentioned this subsequent work in the updated manuscript's discussion section.

*Indeed, fine resolution model simulations are used as the truth. I suggest authors to take advantage of various satellite products to define what scale of heterogeneity needs to be incorporated in each ESM grid cell to better represent land surface states and fluxes.*

The fully distributed model can be seen as the brute-force method that ensures that the multi-scale heterogeneity information in the satellite remote sensing is accounted for in the model. The goal in HMC is to closely approximate the level of detail that would be possible through a fully distributed model with a fraction of the computation. Given that the fully distributed model will also be wrong due to structural and parameter uncertainties, the approach here is meant to be able to find the best of both worlds: allow for large ensemble frameworks to constrain these unavoidable uncertainties while ensuring the robust characterization of the observed multi-scale heterogeneity. That being said, we agree that there needs to be

an assurance that the defined heterogeneity provides a robust representation of the hydrologic heterogeneity and not just simply to reproduce the heterogeneity of the environmental data. Currently, this is accomplished by including land cover information from Landsat in the clustering which is a proxy of the hydrologic controls on the observed heterogeneity of vegetation. However, much more is possible. For example, the clustering approach could take advantage of Sentinel satellite products to provide a more direct representation of the observed heterogeneity of hydrologic response. We have updated the manuscript to discuss this future direction and how it would provide a stronger underlying basis for the derived hydrologic response units.

*Authors have performed a limited sets of sensitivity analysis to assess the impacts of height band length and number of clusters on simulated fluxes. However, no insights have been provided in order to define these parameters in various landscapes. Further discussion will be really helpful to inform the modelers.*

HMC has recently been applied over the globe at a 1-degree grid cell. The required difference in elevation between adjacent height bands ensures a different number of tiles are required per grid cell; grid cells in flat regions need between 5-20 while those in mountainous regions require between 100-400. The subsequent evaluation paper will present these results. However, even this subsequent study uses a predefined set of parameters for HMC. For this purpose, as outlined in the discussion section, future work should explore multiple approaches to determine how to compute the optimal parameters per grid cell. Possible approaches include regionalizing optimized HMC parameters using existing environmental information.

*No physically based approach are presented for delineating the height bands. Can authors implement a more physically based approach for defining the height bands? For example, the methodology of Khan et al. 2013 (Environmental Modelling and Software) for delineating landforms can be informative in this case.*

To ensure the height bands are physically connected to the observed landscape, HMC makes use of the available elevation data to connect each 30-meter grid cell to a corresponding height band. First, the height above nearest drainage area is calculated from the high-resolution elevation data and the computed channel network. The combination of the map of hillslopes and the HAND map are then used to compute the characteristics of each hillslopes (e.g., slope). After clustering the hillslopes using these computed properties including the characteristics profiles, each characteristic hillslope is discretized into $n$ height bands where $n$ is the rounded up value of the HAND value at the ridge for a given characteristic hillslope divided by a user-defined parameter Δh that is the difference between adjacent height bands. These height bands are then connected back to the original map to provide a high-resolution representation as shown in Figure 4. We have improved and clarified the description of this step in HMC in the methods section of the revised manuscript.

*Another issue is regarding the number of variables that are required for a multivariate cluster analysis. I suggest authors to perform a sensitivity analysis to identify factors that are most influential in defining these clusters.*

We agree that the clustering analysis will be susceptible to the number of variables used; if the correlation between many of the variables is high, the additional information provided by the increase in variables will be relatively small. Although a concerted effort was made to ensure the chosen environmental properties were sufficiently independent, further work should be done along this line. One example that is being explored by one of the authors is to use principal component analysis to arrive at the most significant features. Another approach is to use an iterative approach to assign different weights to the different input data. In any case, this is beyond the scope of this study and will be addressed in future research. Section 5.4 in the updated manuscript addresses this question by introducing approaches to more directly cluster on the process heterogeneity which ameliorates some of the challenges of clustering on environmental data that is only partially connected to the modeled biogeochemical cycles.

*Further descriptions about the discretization methodology is required particularly those that relate to Figure 4. I suggest authors to prepare a flowchart that explains every step from discretization to mapping back the results to the fine resolution. Further clarification regarding the simulation elements are required.*

Figures 3 through 5 illustrate the different steps taken in the hierarchical multivariate clustering algorithm. We believe that these figures are sufficiently detailed and do not require further additions. However, we understand the reviewer's concern with the discretization processes. We have expanded the methods section in the revised manuscript for clarification.

*Another major point is how does the surface and subsurface connectivity between the tiles within a single band and between bands are maintained? What rules do you implement?*

The focus of this study is on the development and implementation of the hierarchical multivariate clustering algorithm. For conciseness and simplicity we do not provide a detailed explanation of the hillslope model which is explained in complete detail in [*Subin et al.*, 2014]. For further details, as mentioned in Section 3.1, we encourage readers to read that paper. Here we provide a brief summary of the hillslope model. All tiles that are contained within a characteristic hillslope interact via the subsurface via the exchange of water through Richards' equation. The parameter used for the interaction between height bands is the interface width between two adjacent height bands, the length between the center points of the height bands, and the effective hydraulic conductivity between the two. The exchange of subsurface water occurs horizontally between the different layers of each adjacent height band. A similar approach is used for the intra-height tiles except in this case the interface width is fixed by a user-defined parameter.

Currently, surface runoff is routed instantaneously to the channel at the base of the hillslope.

*How tiles are represented for model simulations? Does each tile represented by a point scale simulation?*

Each tile simulation can be interpreted as a point simulation that is scaled up to the areal coverage of the tile. However, in practice, it is more appropriate to interpret each tile as a field-scale simulation since many of the parameterized processes (e.g., vegetation dynamics) are more suitable for interpretation above ~10 meter scales. These tiles then interact via the subsurface along the discretized hillslope.

*P6-L20: The authors indicate that attributes of each characteristic hillslope are obtained through arithmetic averaging. How do you handle categorical data like soil type and vegetation types in this case?*

Note that categorical data only comes into play in the last step of HMC where soil and land cover data are clustered into intra-band tiles. To cluster this categorical information, these data are split into binary metrics (e.g., grass vs. tree and managed vs. natural). When providing model parameters for LM4-HB some properties such as plant species needs to be defined as a categorical type. In this case, after constructing the map of tiles at 30 meters, the mode of each categorical type for each tile calculated from the 30-meter map of plant species is used as the tile's species. This has been clarified in Section 3.2.3 in the updated manuscript.

*Section 3.2.1. What is a typical size of a characteristic hillslope?*

From our analysis for this domain and other regions, the typical length of each hillslope ranges between 200 to 2000 meters with slopes ranging between 0.01 and 0.5.

*Section 3.2.2. L 30: Needs further explanation.*

We appreciate the reviewer's feedback on the need for clarification within the hillslope discretization section. We have expanded this section in the revised manuscript.

*Can authors provide further insights and general recommendations for implementation of their approach in other geographic regions?*

Ongoing work has used available environmental data to implement HMC over the globe. Section 5.5 in the updated manuscript discusses how future work could explore how to arrive at the optimal HMC parameters per macroscale grid cell over the globe. A subsequent publication will introduce the application of these methods over the globe.

*Can authors provide further information about the enhancements made in this approach compared to the earlier work of delineating HydroBlocks?*

The approach used originally in the HydroBlocks paper [*Chaney et al.*, 2016] can be seen as a "brute force" method that uses multivariate clustering without an explicit accounting of the hydrologic structure. This leads to connections between tiles or HRUs that at times can be unrealistic and lead to poor performance (e.g., subsurface runoff). This paper aims to take the ideas explored in HydroBlocks a step further and make use of common hydrologic knowledge to create a hierarchy of tiles that takes advantage of the clustering approach while ensuring a hydrologically consistent system. As shown in Figure 9, this leads to the convergence of baseflow as the number of tiles increases which is not the case in the HydroBlocks paper.

We would again like to thank both reviewers for their time and helpful comments.

**References**

Chaney, N., P. Metcalfe, and E. F. Wood (2016), HydroBlocks: A Field-scale Resolving Land Surface Model for Application Over Continental Extents, *Hydrol. Process.*, doi:10.1002/hyp.10891.

Subin, Z. M., P. C. D. Milly, B. N. Sulman, S. Malyshev, and E. Shevliakova (2014), Resolving terrestrial ecosystem processes along a subgrid topographic gradient for an earth-system model, *Hydrol. Earth Syst. Sci. Discuss.*, *11*(7).